# A GENERATIVE FRAMEWORK FOR CAUSAL ESTIMATION VIA IMPORTANCE-WEIGHTED DIFFUSION DISTILLATION

## ABSTRACT

Estimating individualized treatment effects from observational data is a central challenge in causal inference, largely due to covariate imbalance and confounding bias from non-randomized treatment assignment. While inverse probability weighting (IPW) is a well-established solution to this problem, its integration into modern deep learning frameworks remains limited. In this work, we propose Importance-Weighted Diffusion Distillation (IWDD), a novel generative framework that combines the pretraining of diffusion models with importance-weighted score distillation to enable accurate and fast causal estimation—including potential outcome prediction and treatment effect estimation. We demonstrate how IPW can be naturally incorporated into the distillation of pretrained diffusion models, and further introduce a randomization-based adjustment that eliminates the need to compute IPW explicitly—thereby simplifying computation and, more importantly, provably reducing the variance of gradient estimates. Empirical results show that IWDD achieves state-of-the-art out-of-sample prediction performance, with the highest win rates compared to other baselines, significantly improving causal estimation and supporting the development of individualized treatment strategies.

## 1 INTRODUCTION

In causal inference, the Neyman–Rubin potential outcomes (PO) framework (Rubin, 2005) formalizes causal effects by comparing potential outcomes under different treatments. The Fundamental Problem of Causal Inference (Holland, 1986) highlights that, for any given unit, only one of the potential outcomes can be observed—the one corresponding to the treatment actually received—while the counterfactual remains unobserved. In randomized controlled trials (RCTs), randomization ensures that treatment assignment is independent of potential outcomes, thus eliminating confounding bias. However, in most observational studies, treatment assignment is typically non-random and may depend on patient-level covariates, leading to covariate imbalance and confounding. This complicates potential outcome estimation and hinders the development of reliable individualized treatment recommendations—particularly in data-scarce settings.

Existing approaches such as inverse probability weighting (IPW) (Robins et al., 1994) address covariate imbalance by reweighting data to approximate RCTs. However, IPW can be unstable due to challenges in propensity score estimation (Liao & Rohde, 2022; Ding, 2023), particularly when propensity scores approach 0 or 1—resulting in extreme weights and high-variance estimators. These issues are further exacerbated when applying IPW to generative models, where propensity networks could be subject to miscalibration and covariate representations may be poorly aligned with treatment assignment (Kallus, 2020). As a result, incorporating IPW in a stable and effective manner into generative frameworks—particularly diffusion-based models—remains an under-addressed challenge.

To address the limitations of existing causal estimation methods, we propose *Importance-Weighted Diffusion Distillation* (IWDD), a novel generative framework for this task. IWDD first pretrains a covariate- and treatment-conditional diffusion model using observational data, then incorporates IPW into its distillation process. An important advantage of this two-stage procedure—diffusion pretraining followed by IPW-modulated distillation—is that pretraining allows the model to fit the

in-sample distribution well, while distillation focuses on learning a conditional generator that adjusts for confounding and covariate imbalance, improving robustness for out-of-sample prediction.

We further show that the IPW-modulated distillation loss can be simplified via a randomization-based adjustment under importance reweighting, eliminating the need to explicitly compute IPW. This not only simplifies implementation but also mitigates approximation bias and numerical instability associated with propensity score estimation. More importantly, the resulting importance-weighted distillation loss is theoretically shown to reduce the variance of gradient estimates, making IWDD a stable and reliable generative approach for causal estimation.

Another inherent benefit of IWDD is its significantly faster sampling speed compared to the pretrained conditional diffusion model. It produces samples in a single forward pass through the network, while the pretrained teacher model requires many iterative refinement steps.

Through extensive empirical studies, we demonstrate that IWDD is an effective approach for training a one-step generator for causal estimation. This establishes IWDD as not only a significantly faster alternative to conditional diffusion models pretrained on observational data, but also a more accurate method for addressing confounding and covariate imbalance inherent in causal inference settings.

We summarize our key contributions as follows:

- We propose IWDD, a novel generative framework for causal estimation that pretrains a conditional diffusion model and distills it into a fast and high-performing one-step generator.
- IWDD is the first to incorporate randomized control adjustment into the distillation process, enabling effective correction for confounding and imbalanced treatment assignment.
- We propose an IPW-modulated diffusion distillation objective and develop an improved variant that removes the need for explicit propensity score estimation. Our theoretical analysis further establishes its risk dominance and reduced gradient variance.
- Empirically, IWDD achieves state-of-the-art performance on multiple benchmark datasets for causal effect estimation, advancing the development of individualized treatment strategies.

## 2 RELATED WORK

**CATE Estimation and PO Prediction.** Estimating the Conditional Average Treatment Effect (CATE) has been extensively studied, with approaches broadly categorized into meta-learners, representation learning, and generative models. Meta-learners such as the S-learner and T-learner (Künzel et al., 2019) recast CATE estimation as a supervised learning problem but are sensitive to covariate imbalance. This limitation has motivated balancing-based methods such as TARNet (Curth & van der Schaar, 2021a;b) and CFR (Shalit et al., 2017). Generative approaches like GANITE (Yoon et al., 2018) further model counterfactual distributions using adversarial training. More recent methods adopt doubly robust strategies, including the DR-learner (Kennedy, 2023) and RA-learner (Curth & van der Schaar, 2021b), which combine nuisance component estimation with pseudo-outcome regression to enhance robustness under limited data. TEDVAE (Zhang et al., 2021) employs variational autoencoders to disentangle latent confounders for treatment effect estimation. While many of these methods support PO prediction, their primary focus is CATE estimation, and they often exhibit limited accuracy when predicting individual-level outcomes.

**Diffusion Models for Causal Estimation.** Recent work has applied diffusion models to causal inference in two major frameworks: the Structural Causal Model (SCM) framework (Pearl, 2009) and the PO framework (Rosenbaum & Rubin, 1983). SCM focuses on modeling causal mechanisms through structural equations and graphs, whereas PO emphasizes treatment assignment and hypothetical interventions, making it particularly well-suited for policy evaluation and randomized experiments (Pearl, 2015). In the SCM setting, diffusion models have been explored for counterfactual generation (Sanchez & Tsaftaris, 2022; Komanduri et al., 2024; Chao et al., 2023; Shimizu, 2023) and causal discovery (Sanchez et al., 2023; Mamaghan et al., 2023; Lorch et al., 2024; Varici et al., 2024). In contrast, diffusion models under the PO framework remain relatively underexplored. DiffPO (Ma et al., 2024) is among the first to use conditional diffusion models to learn potential outcome distributions given covariates and treatment assignments, but it exhibits limitations in its handling of propensity reweighting and sampling efficiency. Our work, also in the PO framework, directly addresses these challenges, aiming to achieve accurate individual-level PO prediction and reliable treatment effect estimation.

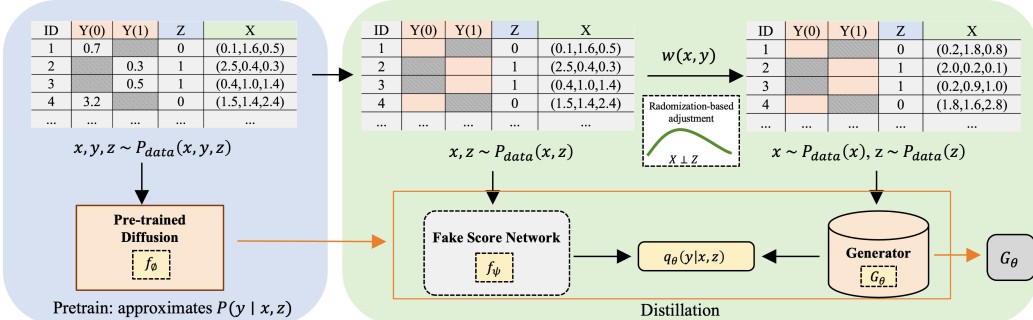

Figure 1: Overview of IWDD. We first pretrain a conditional diffusion model $f_\phi(y \mid x, z)$ on observational data. In distillation, we apply a randomization-based adjustment and train a generator $q_\theta(y \mid x, z)$ using marginal sampling, which implicitly applies importance weighting without requiring explicit propensity estimation. The distillation algorithm is detailed in Algorithm 1.

**Diffusion Distillation.** Diffusion models have been developed to model complex data distributions and enable high-quality sample generation in high-dimensional spaces (Sohl-Dickstein et al., 2015; Song & Ermon, 2019; Ho et al., 2020; Song et al., 2020). Despite their impressive performance across domains, their high computational cost—stemming from the need for hundreds or even thousands of iterative refinement steps—has led to the emergence of diffusion distillation techniques that compress this process into one or a few generation steps.

A foundational strategy in diffusion distillation is to minimize a statistical divergence between the model distribution and the data distribution in the noisy space induced by forward diffusion. While distribution matching in this noisy space was pioneered by Diffusion GAN (Wang et al., 2022; Zheng et al., 2022), it relies on noisy samples to represent the noisy distribution—unlike diffusion distillation methods that leverage pretrained diffusion models to estimate the score of the noisy distribution. A widely adopted divergence in this context is the KL divergence (Poole et al., 2022). Although the KL divergence itself is intractable, its gradient has a tractable form that enables alternating optimization: alternating between estimating the generator's score and updating the generator. Methods that follow this principle include Variational Score Distillation (VSD) (Wang et al., 2023b), Diff-Instruct (Luo et al., 2024), Distribution Matching Distillation (Yin et al., 2024), and their extensions.

Score identity Distillation (SiD) (Zhou et al., 2024) further advances this line of work. By viewing the forward diffusion process through the lens of semi-implicit distributions (Yin & Zhou, 2018; Yu et al., 2023) and leveraging associated score identities (Robbins, 1992; Efron, 2011; Vincent, 2011), SiD replaces the KL divergence with a Fisher divergence and introduces a corresponding alternating optimization procedure. The resulting distillation algorithm achieves one-step generation quality comparable to that of the original pretrained diffusion model after many denoising steps. It does not require access to the original training data to rival the performance of the pretrained teacher, and hence can work well as a data-free method.

## 3    IWDD: IMPORTANCE-WEIGHTED DIFFUSION DISTILLATION

**Notation.** Under the Neyman–Rubin PO framework (Rubin, 2005), we consider an observational dataset $\mathcal{D} = \left\{(X_i, Z_i, Y_i)\right\}_{i=1}^{n}$, where $X \in \mathcal{X} \subseteq \mathbb{R}^d$ denotes covariates, $Z \in \{0, 1\}$ is a binary treatment indicator, and $Y \in \mathcal{Y} \subseteq \mathbb{R}$ is the observed outcome of interest. Let $\pi(x) = P(Z = 1 \mid X = x)$ denote the propensity score, and let $Y(z)$ be the potential outcome under treatment $Z = z$. We denote the observational data distribution as $p_{\text{data}}(x, z, y)$, and the true conditional outcome distribution as $p(y \mid x, z)$. To ensure identifiability of average causal effects from observational data, we adopt the following standard assumptions:

**Assumption 1** (Consistency, Unconfoundedness, and Overlap). *(1) **Consistency:** If individual $i$ receives treatment $Z_i$, then we only observe $Y_i = Y_i(Z_i)$. (2) **Unconfoundedness:** There are no unmeasured confounders, i.e., $\{Y_i(0), Y_i(1)\} \perp Z_i \mid X_i$. (3) **Overlap (Positivity):** Each individual has a non-zero probability of receiving either treatment level; that is, $0 < \pi(x) < 1$ for all $x \in \mathcal{X}$.*

**Problem Formulation.** When using a generative model for causal estimation, given observational data $\{x_i, z_i, y_i\}_{i=1}^n$, one can pretrain a covariate- and treatment-conditional diffusion model and use the reverse diffusion process to approximate $p(y \mid x, z)$, conditioned on the input $(x, z)$. This model can then be directly applied for generative causal estimation. We show that this approach serves as a strong baseline for in-sample causal estimation. However, not only is it slow to generate random samples of $y$ given $(x, z)$, but its performance also noticeably degrades in out-of-sample settings where $(x, z)$ lie in low-density regions of the training data (Li et al., 2025).

To eliminate confounding effects and mitigate issues arising from the imbalanced distribution of treatment assignments $z$ in the training dataset, we propose a diffusion distillation-based framework that incorporates the principles of RCTs into the distillation process. The ultimate goal of this framework is to learn a generative distribution $q_\theta(y \mid x, z)$ that effectively accounts for confounding and covariate-treatment imbalance, which are common challenges in real-world observational data. We define this distribution implicitly via its generation process:

$$y_g = G_\theta(x, z, \varepsilon), \quad \varepsilon \sim \mathcal{N}(0, \mathbf{I}), \tag{1}$$

where $G_\theta$ is a deep neural network-based one-step generator parameterized by $\theta$. Causal estimation is then conducted by generating samples $y_g$ from this model, conditioned on both the covariates $x$ and the treatment assignment $z$. In what follows, we detail the construction of IWDD as an effective approach for training $G_\theta$, which outperforms the pretrained diffusion model in estimating the conditional distribution of $y$ given $x$ and $z$, particularly when $(x, z)$ lie in low data density regions.

## 3.1 PRETRAINING OF COVARIATE- AND TREATMENT-CONDITIONAL DIFFUSION MODELS

We begin by fitting a covariate- and treatment-conditional diffusion model (Sohl-Dickstein et al., 2015; Ho et al., 2020; Han et al., 2022) $f_\phi(y \mid x, z)$, parameterized by $\phi$, to approximate the true conditional distribution $p(y \mid x, z)$ over observational data $(x, z, y) \sim p_{\text{data}}(x, z, y)$:

Given a data point $(y_0, x, z)$ from $p_{\text{data}}(y, x, z)$, in the forward process, Gaussian noise is gradually added to the initial outcome $y_0$ over $T$ discrete time steps. This produces a sequence of progressively noisier samples $y_1, \ldots, y_T$. The forward process is defined as: $q(y_t \mid y_0) = \mathcal{N}(a_t y_0, \sigma_t^2 I)$, with $a_t \in [0, 1]$. To generate $y_t$ given $y_0$, we apply the standard reparameterization: $y_t = a_t y_0 + \sigma_t \epsilon_t, \epsilon_t \sim \mathcal{N}(0, I)$. We follow the EDM schedule of Karras et al. (2022), setting $a_t = 1$ while adjusting $\sigma_t$ (details in Appendix E.1). To learn the reverse process, we train the conditional denoising function $f_\phi$ using the following objective:

$$\mathcal{L}_\phi = \mathbb{E}_{\sigma, (y, x, z), n} \left[ \lambda(\sigma) \| f_\phi(y_t; \sigma, x, z) - y \|_2^2 \right].$$

A well-trained teacher diffusion model $f_\phi$ is capable of estimating $\mathbb{E}[y \mid y_t, x, z]$. It serves as both the teacher and the initialization for the subsequent adjusted distillation.

DiffPO (Ma et al., 2024), a recent baseline, incorporates inverse propensity score reweighting into this diffusion loss to adjust for confounding between $x$ and $z$:

$$\mathbb{E}_{(y_0, x, z) \sim p(y, x, z), \, \epsilon \sim \mathcal{N}(0, I), \, t} \left[ w(x, z) \| \epsilon - \epsilon_\phi(y_0 + \sigma_t \epsilon, t \mid x, z) \|^2 \right]. \tag{2}$$

where $w(x, z) = \frac{1}{p(z|x)} = \frac{z}{\pi(x)} + \frac{1-z}{1-\pi(x)}$, and $\pi(x) = p(z = 1 \mid x)$ denotes the propensity score.

## 3.2 DISTILLATION VIA IMPORTANCE REWEIGHTING

Unlike training a diffusion model, which requires multiple reverse steps for sampling, our goal is to train a one-step conditional generator $q_\theta(y \mid x, z)$ that approximates the true conditional distribution $p(y \mid x, z)$ for all $(x, z) \in \mathcal{X} \times \mathcal{Z}$. In observational data, however, samples $(x, z) \sim p_{\text{data}}(x, z)$ are typically *not* drawn independently across covariates and treatment. That is, the treatment assignment $z$ may depend on covariates $x$, inducing imbalance across treatment groups. Consequently, a model that optimizes a vanilla divergence $\mathbb{E}_{p_{\text{data}}(x,z)}[D(q_\theta(y \mid x, z), p(y \mid x, z))]$ can bias distillation toward regions where $(x, z)$ pair occurs more frequently. In practice, this can lead to better performance on the majority treatment group while degrading generalization in underrepresented regions, as demonstrated in our synthetic example in Section 4.1.

**IPW-based importance weighting.** To correct for the sampling bias arising from the non-random treatment assignment in the observed joint distribution $p_{\text{data}}(x, z)$, we apply an importance weighting

factor based on the discrepancy between $p_{\text{data}}(x, z)$ and the product of marginals $p_{\text{data}}(x)p_{\text{rct}}(z)$, where $p_{\text{rct}}(z) = \text{Bernoulli}(z; 0.5)$ reflects the ideal joint distribution of $x$ and $z$ under RCTs. We reweigh every sample by:

$$w(x, z) = \frac{p_{\text{data}}(x)p_{\text{rct}}(z)}{p_{\text{data}}(x, z)} = \frac{p_{\text{rct}}(z)}{p_{\text{data}}(z \mid x)}, \tag{3}$$

where $p_{\text{data}}(z = 1 \mid x) \equiv \pi(x)$ is the *propensity score* (Rosenbaum & Rubin, 1983). This leads to an inverse-propensity weighted divergence loss:

$$\mathcal{L}_\theta^{\text{IPW}} = \mathbb{E}_{(x,z) \sim p_{\text{data}}(x,z)} \left[ w(x, z) \cdot D \left( q_\theta(y \mid x, z), p(y \mid x, z) \right) \right], \tag{4}$$

where the divergence $D(q_\theta, p)$ is defined as $D(q_\theta, p) = \mathbb{E}_{y \sim q}[d(q_\theta, p)]$, for some pointwise divergence measure $d(q_\theta, p)$ (see Section 3.2.2 for the specific choice of divergence we adopt).

Recent works (Ma et al., 2024; Mahajan et al., 2024) also use the inverse-propensity weights $1/p_{\text{data}}(z \mid x)$. They train a propensity network $g_\omega$ to obtain $\hat{\pi}(x) = g_\omega(x)$ and form the weights $1/\hat{\pi}(x)$ (or $1/[1 - \hat{\pi}(x)]$). When $p_{\text{data}}(z \mid x)$ approaches zero, this leads to an excessively large weight which causes numerical instability. Common approaches to improve stability include: (i) *truncating* the estimated propensity scores to a fixed interval, and (ii) *trimming* the sample by discarding units with propensity scores outside that interval. Although both stabilize the IPW estimators, they introduce additional arbitrariness (Ding, 2023). Moreover, improper clipping risks nullifying the intended reweighting effect. We identified such an issue in DiffPO (Ma et al., 2024). Although it performs well on some datasets, its implementation incorrectly truncates $1/p_{\text{data}}(z \mid x)$ to values below one—despite the fact that $1/p_{\text{data}}(z \mid x) \geq 1$ by definition.[1] This improper implementation nullifies the intended effect of propensity score reweighting, effectively reducing the objective in Equation 2 to a standard diffusion loss without any reweighting.

**Implicit importance weighting via marginal sampling.** Given the pitfalls of existing IPW-based importance weighting, we now introduce a key result showing that the importance reweighting objective in Equation 4 can be reparameterized without explicitly computing weights through training a neural network for propensity score.

**Lemma 1.** *The importance-weighted loss in Equation 4 is equivalent to the expected divergence under the product of marginals:*

$$\mathcal{L}_\theta^{\text{IWDD}} = \mathbb{E}_{(x,z) \sim p_{\text{data}}(x)p_{\text{rct}}(z)} \left[ D \left( q_\theta(y \mid x, z), p(y \mid x, z) \right) \right]. \tag{5}$$

*Proof.* Substituting the importance weight $w(x, z) = \frac{p_{\text{data}}(x)p_{\text{rct}}(z)}{p_{\text{data}}(x,z)}$ into Equation 4, we have:

$$\mathcal{L}_\theta^{\text{IPW}} = \mathbb{E}_{(x,z) \sim p_{\text{data}}(x,z)} \left[ w(x, z) \cdot D(q_\theta(y \mid x, z), p(y \mid x, z)) \right]$$
$$= \mathbb{E}_{(x,z) \sim p_{\text{data}}(x)p_{\text{rct}}(z)} \left[ D(q_\theta(y \mid x, z), p(y \mid x, z)) \right] = \mathcal{L}_\theta^{\text{IWDD}}. \qquad \square$$

This equivalence enables us to apply the bias correction implicitly through a sampling adjustment: we sample $x \sim p_{\text{data}}(x)$ and independently draw $z \sim \text{Bernoulli}(0.5)$. This yields the importance weight $w(x, z) = 1/p(z \mid x)$. This approach bypasses the need for propensity score estimation or weight clipping, as $w(x, z) = 1/p(z \mid x)$ is never explicitly computed. We will use the loss $\mathcal{L}_\theta^{\text{IWDD}}$ as the generator loss in Algorithm 1.

**Gradient variance advantage.** Although Lemma 1 shows that $\mathcal{L}_\theta^{\text{IWDD}}$ and $\mathcal{L}_\theta^{\text{IPW}}$ have the same expectation, gradient estimates under $\mathcal{L}_\theta^{\text{IWDD}}$ can exhibit lower variance compared to those under $\mathcal{L}_\theta^{\text{IPW}}$ when $w(x, z) = p_{\text{rct}}(z)/p_{\text{data}}(z \mid x)$ (see Theorem 2 in Appendix A). By removing explicit dependence on inverse propensity scores, the IWDD formulation in Equation 5 achieves **lower gradient variance** than the IPW-based objective in Equation 4, avoiding instability from high-variance weights, and thereby leading to more efficient optimization with faster, more stable convergence.

### 3.2.1 Radomization-based Adjustment for Distillation

When sampling from $p_{\text{data}}(x)$ and $p_{\text{rct}}(z)$, we propose a novel randomization-based adjustment to the data used for training the generator $q_\theta$, improving performance over standard distillation. Since RCTs

---

[1]See DiffPO's official implementation at commit `43ebb60`: `https://github.com/yccm/DiffPO/blob/43ebb6048dc09b0315e8f25db9b5d00a95b9b3e0/src/main_model.py#L144-L147`

are the gold standard for estimating treatment effects, our goal is to approximate an RCT-like setting by breaking the dependence between covariates X and treatment assignment Z. Specifically, randomly **shuffling** $X$ can eliminate existing associations between $X$ and $Z$ while preserving their marginal distributions. However, to ensure a balanced treatment assignment across individuals, we instead **sample** $Z$ independently from a Bernoulli$(0.5)$ distribution. This setup aligns with our practice of predicting both $Y_0$ and $Y_1$ for each individual, effectively mirroring the conditions of an RCT.

While past literature has emphasized the importance of randomized designs and balancing methods to reduce bias in observational studies (Imbens & Rubin, 2015; Rosenbaum & Rubin, 1983; Stuart, 2010), our adjustment represents a novel randomization procedure specifically tailored to diffusion distillation-based causal estimation.

### 3.2.2 CHOICE OF DIVERGENCE

We consider two representative divergence measures: the KL divergence and the Fisher divergence. Under both, the gradient of the divergence loss $\mathcal{L}_\theta$ can be estimated via an alternating optimization procedure between a fake score network and a generator. The fake score network is trained to approximate the score of the generated response variable $y$ given $x, z \sim p_{\text{data}}(x, z)$, while the generator is trained to optimize $q_\theta(y \mid x, z)$, which will ultimately be used for causal estimation.

We implemented KL divergence-based distillation following prior work (Wang et al., 2023a; Luo et al., 2023; Yin et al., 2024). While this approach improves sampling efficiency, it fails to improve upon the original model and often performs worse, as shown in Row 2 of Figure 3. Moreover, it exhibits training instability and is prone to collapse in our synthetic data experiments. In contrast, a Fisher divergence objective combined with SiD-based gradient estimation (Zhou et al., 2024; Chen et al., 2025) results in more stable training and consistently stronger empirical performance. Thus, we adopt Fisher divergence as the distillation objective in IWDD and use SiD to optimize it, leading to a one-step generator that achieves both high sampling efficiency and strong causal estimation accuracy.

During distillation, we alternate between updating the generator $G_\theta$ and the fake score network $f_\psi$. The generator $G_\theta$ is trained on randomized pairs $(\tilde{x}, \tilde{z})$ obtained via randomization-based adjustment to distill a pretrained diffusion model $f_\phi$, originally trained on the joint distribution $p_{\text{data}}(y, x, z)$. The fake score network is trained on unadjusted observational pairs $(x, z) \sim p_{\text{data}}(x, z)$ to approximate the score of the generator's output distribution. By applying randomization-based adjustment only to the generator inputs while keeping the fake score network conditioned on observational data, we generalize the approach of Zhou et al. (2024) and define the two loss functions as follows:

Generator loss:  $\mathcal{L}_\theta = w(t) \, (f_\phi(y_t \mid \tilde{x}, \tilde{z}) - f_\psi(y_t \mid \tilde{x}, \tilde{z}))^\top \, (f_\psi(y_t \mid \tilde{x}, \tilde{z}) - y_g)$

$$+ (1 - \alpha) \, w(t) \, \|f_\phi(y_t \mid \tilde{x}, \tilde{z}) - f_\psi(y_t \mid \tilde{x}, \tilde{z})\|_2^2, \quad \text{where } (\tilde{x}, \tilde{z}) \sim p_{\text{data}}(\tilde{x}) p_{\text{rct}}(\tilde{z}). \quad (6)$$

Fake loss:   $\mathcal{L}_\psi = \gamma(t) \, \|f_\psi(y_t \mid x, z) - y_g\|_2^2, \quad \text{where } (x, z) \sim p_{\text{data}}(x, z). \quad (7)$

Here, $y_g$ is the sample generated by the one-step generator $G_\theta$, and $y_t = y_g + \sigma_t \epsilon_t$; $f_\phi$, the teacher diffusion model, is pretrained to estimate $\mathbb{E}[y_0 \mid y_t, x, z]$ and kept frozen during distillation; $f_\psi$, the fake score network, is trained to match $\mathbb{E}[y_g \mid y_t, x, z]$. The full algorithm is in Algorithm 1. Detailed training schedules and the weighting functions $w(t)$ and $\gamma(t)$ are provided in Appendix E.1.

---

**Algorithm 1** IWDD Training

---

**Require:** Pretrained diffusion $f_\phi$, training data $\mathcal{D} = \{(x_i, z_i)\}_{i=1}^n$, batch size $B$
1: **Initialize:** $\theta \leftarrow \phi$, $\psi \leftarrow \phi$
2: **repeat**
3:     Sample mini-batch indices $\mathcal{I} \subset \{1, \ldots, n\}$ with $|\mathcal{I}| = B$
4:     $x \leftarrow \{x_i\}_{i \in \mathcal{I}}; \quad z \leftarrow \{z_i\}_{i \in \mathcal{I}}$
5:     $\tilde{x} \leftarrow \texttt{shuffle}(x); \quad \tilde{z} \sim \text{Bernoulli}(0.5)$            ▷ Randomization-based adjustment
6:     $y_g \leftarrow G_\theta(\tilde{x}, \tilde{z}, \varepsilon), \ \varepsilon \sim \mathcal{N}(0, \mathbf{I})$, and let $y_t = y_g + \sigma_t \epsilon_t, \ \epsilon_t \sim \mathcal{N}(0, \mathbf{I})$
7:     $\theta \leftarrow \theta - \eta_\theta \nabla_\theta \mathcal{L}_\theta(y_g, \epsilon_t, \tilde{x}, \tilde{z})$            ▷ Equation 6
8:     $\psi \leftarrow \psi - \eta_\psi \nabla_\psi \mathcal{L}_\psi(y_g, \epsilon_t, x, z)$            ▷ Equation 7
9: **until** Converge
**Ensure:** Trained generator $G_\theta$

---

## 3.3 RISK DOMINANCE OF DISTILLATION

We establish that distillation not only accelerates the sampling of teacher diffusion but also reduces its estimation risk, leading to more accurate predictions (see Section 4 for empirical results). Theoretically, we show the one-step generator of IWDD, $G_\theta$, achieves performance that is never worse, and generally strictly better, on the target RCT distribution compared to the pretrained model $f_\phi$.

Let $\mathcal{G} \subseteq \{g : \mathcal{X} \times \{0, 1\} \to \mathbb{R}^d\}$ be the hypothesis class. Define the risk functionals

$$R_{\text{train}}(g) = \mathbb{E}_{p_{\text{train}}}\left[\|y - g(x, z)\|_2^2\right], \qquad R_{\text{train}}^{(w)}(g) = \mathbb{E}_{p_{\text{train}}}\left[w(x, z)\|y - g(x, z)\|_2^2\right],$$

with weights $w(x, z) = \frac{p_{\text{RCT}}(z)\, p(x)}{p_{\text{train}}(x, z)} > 0$. By construction, the pretrained diffusion model satisfies $f_\phi \in \arg\min_{g \in \mathcal{G}} R_{\text{train}}(g)$, while the distilled generator satisfies $G_\theta \in \arg\min_{g \in \mathcal{G}} R_{\text{train}}^{(w)}(g)$.

**Theorem 1** (Risk dominance of the generator). *Suppose (i) distributional shift occurs only in $(x, z)$, with invariant outcome mechanism: $p_{\text{test}}(x, z) = p(x)\, p_{\text{RCT}}(z)$ and $p_{\text{test}}(y \mid x, z) = p_{\text{train}}(y \mid x, z)$; and (ii) overlap: $0 < w(x, z) \leq W < \infty$ $p_{\text{train}}$-a.s. Then*

$$R_{\text{test}}(G_\theta) \;\leq\; R_{\text{test}}(f_\phi).$$

*Strict inequality holds whenever $f_\phi$ does not minimize $R_{\text{test}}$ over $\mathcal{G}$; equivalently, see Appendix B.*

*Proof sketch.* By change of measure and the invariance of $p(y \mid x, z)$,

$$R_{\text{train}}^{(w)}(g) = \mathbb{E}_{p_{\text{train}}}\left[w(x, z)\, \|y - g(x, z)\|_2^2\right] = \mathbb{E}_{p_{\text{test}}}\left[\|y - g(x, z)\|_2^2\right] = R_{\text{test}}(g).$$

Hence the minimizers of $R_{\text{train}}^{(w)}$ and $R_{\text{test}}$ over $\mathcal{G}$ coincide, so $R_{\text{test}}(G_\theta) \leq R_{\text{test}}(g)$ for all $g \in \mathcal{G}$, in particular $g = f_\phi$. If $f_\phi$ is not a minimizer of $R_{\text{test}}$, there exists $\bar{g}$ with $R_{\text{test}}(\bar{g}) < R_{\text{test}}(f_\phi)$, which implies $R_{\text{test}}(G_\theta) < R_{\text{test}}(f_\phi)$. A detailed proof is in Appendix B. $\qquad\square$

## 4 EXPERIMENTS

We evaluate IWDD on both synthetic and benchmark datasets. First, we design a synthetic study to highlight IWDD's robustness under covariate shift, including violations of the overlap assumption. Second, we benchmark IWDD on standard causal inference datasets, comparing its performance against baselines in potential outcome prediction and heterogeneous treatment effect estimation.

**Performance metrics.** We evaluate model performance on estimation accuracy for both PO prediction and treatment effects estimation. PO prediction is measured by **Root Mean Squared Error (RMSE)**, where lower values indicate better accuracy. Treatment effect estimation is measured by **Precision in Estimation of Heterogeneous Effect (PEHE)**, $\epsilon_{\text{PEHE}}$. We also report win rates, i.e., the percentage of times a method outperforms others. Formal definitions of all metrics are provided in Appendix F.

### 4.1 SYNTHETIC DATA EXAMPLE: OUT-OF-SAMPLE GENERALIZATION WITHOUT OVERLAP

We demonstrate IWDD's robustness under extreme covariate shift, even to the extent of violating overlap assumption, a challenge both theoretically important and practically common. In some observational studies, treatment is assigned deterministically based on covariates. Verifying whether overlap holds in practice can be difficult, and recent research has emphasized the need for methods that remain effective when overlap fails (Cai et al., 2025; Jin et al., 2025).

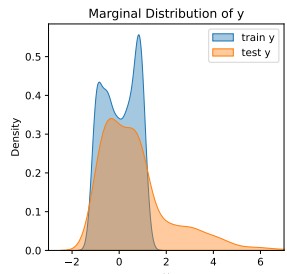

Figure 2: Marginal distributions of $y$ in training vs. testing, differing due to treatment policy shift.

Although the overlap assumption is imposed in earlier sections for identification and theoretical analysis, real-world settings frequently violate it. To stress-test IWDD under this scenario, we construct a synthetic experiment. The covariate distribution $p(x) \sim \mathcal{N}(0, 1)$ and outcome model $y = f(x, z) + \epsilon, \epsilon \sim \mathcal{N}(0, 1)$ are fixed across training and testing. The treatment assignment differs: in training, $z = \mathbf{1}\{x < -1\}$, yielding only 16% treated samples; in testing, $z \sim \text{Bernoulli}(0.5)$. Note that in testing, $z = 1$ can occur when $x \geq -1$, creating $(x, z)$ pairs unseen during training and contributing

to the right tail of the marginal y distribution in Figure 2. This shift in $p(z \mid x)$ induces covariate shift, altering the marginal distribution of $y$. We evaluate performance in two settings: in-sample (using the training distribution $p_{\text{data}}(x, z)$) and out-of-sample (under the test distribution $p_{\text{data}}(x)p_{\text{rct}}(z)$). Specific data-generating mechanism is in Appendix C.

We denote the POs as $Y(0) = f(x, 0) + \epsilon$ and $Y(1) = f(x, 1) + \epsilon$, representing the untreated and treated outcomes. Figure 3 shows that the pretrained diffusion model (Row 1) performs well in-sample and on out-of-sample $Y(0)$. However, it struggles with estimating out-of-sample $Y(1)$ due to the limited treated samples and unseen $(x, z)$ pairs in the training distribution, with an RMSE of 3.07. Row 2 illustrates the KL divergence–based distillation failing to improve upon the pretrained model, consistent with Section 3.2.2. IWDD (Row 3) improves estimation, with noticeable gain in the out-of-sample $Y(1)$ predictions, reducing RMSE from 3.07 to 2.76. This demonstrates its effectiveness in addressing treatment imbalance and enhancing generalization under covariate shift.

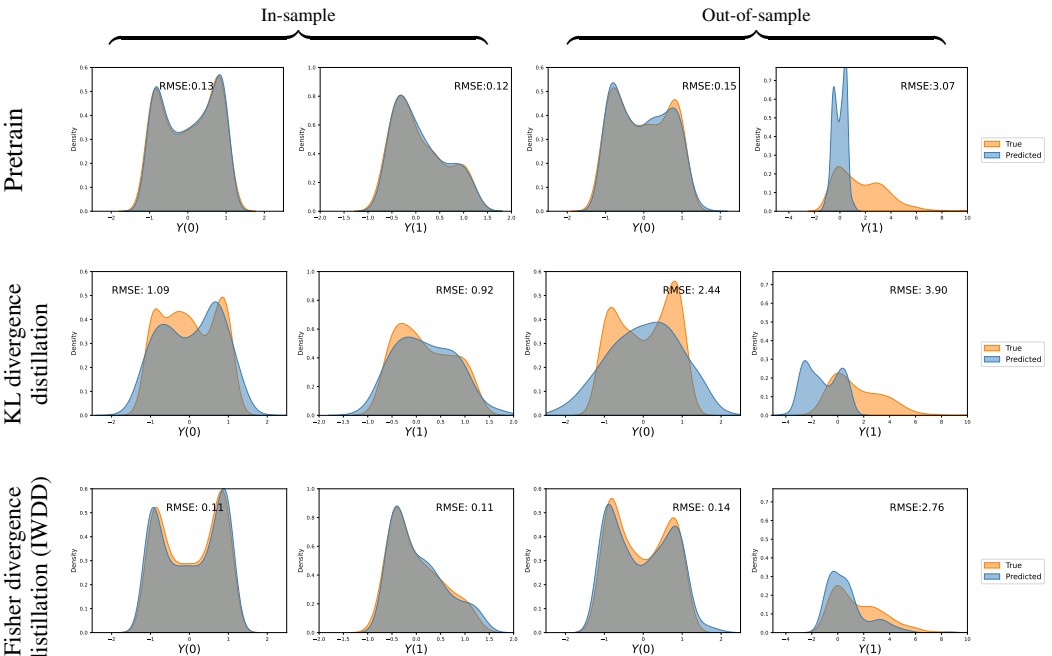

Figure 3: Synthetic data example: estimated potential outcome distributions $Y(0)$ and $Y(1)$ from different models. The pretrained diffusion model performs well in-sample and for $Y(0)$ out-of-sample, but struggles with $Y(1)$. IWDD improves estimation for out-of-sample $Y(1)$ while maintaining performance elsewhere.

The treatment assignment is deterministic (no overlap), so causal effects are unidentifiable from observational data and standard weighting estimators are ill-defined. IWDD does not bypass this identification barrier; rather, it remains well-posed by minimizing the expected divergence to the true conditional outcome distribution under the randomized test policy (the product of marginals). This objective targets the evaluation distribution without relying on overlap. In Appendix H, we show that if the generator class contains a model closer to the target distribution than the pretrained diffusion model, then an approximately optimized IWDD generator will provably yield a strictly better approximation, even without overlap. Empirically, the pretrained model performs relatively poorly on unseen $Y(1)$, while IWDD reduces target error because it (i) trains under the correct policy (RCT) and (ii) projects predictions into a hypothesis class that better fits the target distribution.

## 4.2 BENCHMARKING ON STANDARD PUBLIC DATASETS

**Datasets.** We evaluate our models on three widely used causal inference benchmarks. The **ACIC 2016** dataset includes 77 semi-synthetic datasets generated from real-world health care covariates, with 4802 observations and 55 covariates.[2] The **ACIC 2018** dataset consists of 12 semi-synthetic

---

[2] https://jenniferhill7.wixsite.com/acic-2016/competition

datasets with 10000 observations and 177 covariates.[3] The **IHDP** (Gross, 2024) dataset contains 747 units and 25 covariates, and is based on a real randomized trial with simulated outcomes.

**Baselines.** In our evaluations, we compare against S-learner and T-learner (Künzel et al., 2019), TARNet and CFR (Curth & van der Schaar, 2021b), GANITE (Yoon et al., 2018), and DiffPO (Ma et al., 2024) for potential outcome prediction, and further include DR-learner (Kennedy, 2023), RA-learner (Curth & van der Schaar, 2021b), and TEDVAE (Zhang et al., 2021) for CATE estimation.

**Results and Discussion.** Tables 1 and 2 summarize ACIC 2018 results; full per-dataset results are in Appendix I.2. Additional experiments on ACIC 2016 and IHDP appear in Appendices I.1 and I.3.

Across all benchmarks, IWDD consistently achieves the **best out-of-sample accuracy** for both PO prediction and CATE estimation, highlighting its strong generalization ability. While OffsetNet and FlexTENet (Curth & van der Schaar, 2021a) perform competitively in-sample, their architectures (hard reparametrization in OffsetNet and flexible multi-task subspaces in FlexTENet) tend to overfit the training distribution, leading to poor out-of-sample generalization. IWDD also exhibits **numerical stability** across datasets, whereas other baselines exhibit varying degrees of instability; in particular, DiffPO is unstable despite propensity clipping due to its DDPM-based schedules. Notably, our pretrained diffusion baseline is already stable, but IWDD further improves upon it, attaining the best overall accuracy, consistent with our theoretical analysis of risk dominance in Section 3.3. Further discussion of baseline behaviors is provided in Appendix G.

Table 1: Win rates (%) [4] and mean RMSE for $Y(0)$, $Y(1)$ across 12 ACIC 2018 datasets. The best result across all methods is highlighted in **bold**, and the best result among the diffusion-based approaches (DiffPO, Pretrain, IWDD) is additionally marked with a $\star$.

| Method | In-sample | | | | Out-of-sample | | | |
|---|---|---|---|---|---|---|---|---|
| | $\text{Win}_0(\%)$ | $\text{RMSE}_0$ | $\text{Win}_1(\%)$ | $\text{RMSE}_1$ | $\text{Win}_0(\%)$ | $\text{RMSE}_0$ | $\text{Win}_1(\%)$ | $\text{RMSE}_1$ |
| T-learner | 0 | 65.9 | 8.3 | 68.2 | 0 | 68.2 | 8.3 | 307.3 |
| S-learner | 8.3 | 64.5 | 0 | 65.7 | 8.3 | 275.9 | 0 | 306.8 |
| TNet | 0 | 0.857 | 0 | 0.973 | 8.3 | 1.018 | 0 | 1.120 |
| TARNet | 0 | 0.855 | 0 | 0.967 | 0 | 1.022 | 0 | 1.113 |
| OffsetNet | **75** | **0.786** | **33.3** | **0.889** | 0 | 1.114 | 0 | 1.218 |
| FlexTENet | 8.3 | 0.876 | 16.7 | 0.987 | 0 | 1.058 | 0 | 1.149 |
| DiffPO | 8.3$\star$ | 473.9 | 25$\star$ | 975.2 | 16.7 | 472.7 | 25 | 975.2 |
| Pretrain | 0 | 0.989 | 0 | 1.170 | 0 | 0.902 | 0 | 1.085 |
| **IWDD** | 0 | 0.963$\star$ | 16.7 | 1.038$\star$ | **66.7** | **0.880**$\star$ | **75** | **0.950**$\star$ |

## 5 DISCUSSION

This work introduces IWDD, a generative causal estimation framework that integrates diffusion models, importance weighting, and distillation to address covariate imbalance while enabling efficient one-step sampling. Empirical results on synthetic and real-world datasets demonstrate its robustness and strong generalization. Despite its strengths, IWDD has several limitations. It assumes no unmeasured confounding, a strong assumption, and has so far been evaluated primarily on benchmark datasets. Future work will extend IWDD to settings including complex treatments, discrete outcomes, and longitudinal data, while also exploring ways to further relax identification assumptions and improve training efficiency.

Table 2: Win rates (%) and mean $\epsilon_{\text{PEHE}}$ on ACIC 2018

| | In-sample | | Out-of-sample | |
|---|---|---|---|---|
| | Win(%) | $\epsilon_{\text{PEHE}}$ | Win(%) | $\epsilon_{\text{PEHE}}$ |
| Causal Forest | 0% | 13.452 | 8% | 18.190 |
| T-learner | 8% | 15.175 | 8% | 18.724 |
| S-learner | 0% | 8.613 | 0% | 13.619 |
| TNet | 0% | 0.600 | 0% | 0.574 |
| TARNet | 0% | 0.618 | 0% | 0.619 |
| OffsetNet | 0% | 0.606 | 0% | 0.606 |
| FlexTENet | 0% | 0.652 | 0% | 0.646 |
| DRNet | 0% | 0.607 | 0% | 0.606 |
| DiffPO | 33% | 583.4 | 33% | 589.1 |
| Pretrain | 25% | 0.318 | 33% | 0.306 |
| **IWDD** | **50%** | **0.308** | **58%** | **0.301** |

exploring ways to further relax identification assumptions and improve training efficiency.

---

[3] https://www.synapse.org/#!Synapse:syn11294478/wiki/

[4] Ties are counted for both methods when calculating win rates.

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

# A    GRADIENT VARIANCE COMPARISON FOR IPW-BASED AND IWDD LOSSES

**Theorem 2.** *Let* $\mathrm{Var}_{\mathrm{IWDD}}$ *and* $\mathrm{Var}_{\mathrm{IPW}}$ *denote the gradient covariance matrices under the IWDD loss (Equation 5) and the IPW loss (Equation 4), respectively. Then, the gradient variance under marginal sampling (IWDD) is upper bounded by that of the importance-weighted approach:*

$$\mathrm{Var}_{\mathrm{IWDD}} \preceq \mathrm{Var}_{\mathrm{IPW}}.$$

*Proof.* We have

$$\mathcal{L}_\theta^{\mathrm{IPW}} = \mathbb{E}_{(x,z) \sim p_{\mathrm{data}}(x,z)} \left[ w(x,z) \cdot D\left( q_\theta(y \mid x,z), \, p(y \mid x,z) \right) \right], \tag{8}$$

and

$$\mathcal{L}_\theta^{\mathrm{IWDD}} = \mathbb{E}_{(x,z) \sim p_{\mathrm{data}}(x) p_{\mathrm{rct}}(z)} \left[ D\left( q_\theta(y \mid x,z), \, p(y \mid x,z) \right) \right]. \tag{9}$$

Let

$$g(x,z) \;=\; \nabla_\theta \, D\big( q_\theta(y \mid x,z), p(y \mid x,z) \big),$$

and the importance weight

$$w(x,z) \;=\; \frac{p_{\mathrm{rct}}(z)}{p_{\mathrm{data}}(z \mid x)} \;=\; \begin{cases} \dfrac{1/2}{\pi(x)} & z = 1, \\[2mm] \dfrac{1/2}{1 - \pi(x)} & z = 0. \end{cases}$$

with $\pi(x) := p_{\mathrm{data}}(z = 1 \mid x)$ denoting the propensity score, which by Assumption 1 satisfies $0 < \pi(x) < 1$. We compare two stochastic-gradient estimators:

1. Importance-weighted using propensity score (sampling $(x,z) \sim p_{\mathrm{data}}(x,z)$, then weighting):

$$\hat{g}_{\mathrm{IPW}}(x,z) = w(x,z)\, g(x,z).$$

2. Marginal-sampling (sampling $(x,z) \sim p_{\mathrm{data}}(x)\, p_{\mathrm{rct}}(z)$, no weight):

$$\hat{g}_{\mathrm{IWDD}}(x,z) = g(x,z).$$

In both cases the population gradient is

$$G \;=\; \nabla_\theta \mathcal{L}_\theta = \mathbb{E}_{p_{\mathrm{data}}(x,z)}[\, w\, g\,] = \mathbb{E}_{p_{\mathrm{data}}(x)\, p_{\mathrm{rct}}(z)}[\, g\,].$$

Variance under importance weighting:

$$\mathrm{Var}_{\mathrm{IPW}} = \mathbb{E}_{p_{\mathrm{data}}(x,z)}\big[\hat{g}_{\mathrm{IPW}}\, \hat{g}_{\mathrm{IPW}}^\top\big] \;-\; G\, G^\top$$
$$= \mathbb{E}_{p_{\mathrm{data}}(x,z)}\big[w(x,z)^2\, g(x,z)\, g(x,z)^\top\big] \;-\; \Big(\mathbb{E}_{p_{\mathrm{data}}(x,z)}\big[w(x,z)\, g(x,z)\big]\Big)\Big(\mathbb{E}_{p_{\mathrm{data}}(x,z)}\big[w(x,z)\, g(x,z)\big]\Big)^\top.$$

Variance under marginal sampling:

Since for any test function $h$,

$$\mathbb{E}_{p_{\mathrm{data}}(x)\, p_{\mathrm{rct}}(z)}[h(x,z)] = \mathbb{E}_{p_{\mathrm{data}}(x,z)}\big[w(x,z)\, h(x,z)\big],$$

we have

$$\mathrm{Var}_{\mathrm{IWDD}} = \mathbb{E}_{p_{\mathrm{data}}(x)\, p_{\mathrm{rct}}(z)}\big[g(x,z)\, g(x,z)^\top\big] \;-\; G\, G^\top$$
$$= \mathbb{E}_{p_{\mathrm{data}}(x,z)}\big[w(x,z)\, g(x,z)\, g(x,z)^\top\big] \;-\; \Big(\mathbb{E}_{p_{\mathrm{data}}(x,z)}\big[w(x,z)\, g(x,z)\big]\Big)\Big(\mathbb{E}_{p_{\mathrm{data}}(x,z)}\big[w(x,z)\, g(x,z)\big]\Big)^\top.$$

Comparing the two variances:

$$\mathrm{Var}_{\mathrm{IPW}} - \mathrm{Var}_{\mathrm{IWDD}} = \mathbb{E}_{p_{\mathrm{data}}(x,z)}\big[(w^2 - w)\, g\, g^\top\big], \tag{10}$$

Condition on $x$ and write $\Sigma(x) = \sum_z p_{\text{data}}(z \mid x) \, g(x,z) g(x,z)^\top \succeq 0$. With $\pi(x) = p_{\text{data}}(z = 1 \mid x)$ abbreviating the propensity score, the inner expectation in Equation 10 becomes

$$\Delta(x) := \sum_{z=0}^{1} p_{\text{data}}(z \mid x) \left( w^2(x,z) - w(x,z) \right)$$

$$= \pi(x) \left[ \left( \tfrac{1}{2}/\pi(x) \right)^2 - \tfrac{1}{2}/\pi(x) \right] + \left[ 1 - \pi(x) \right] \left[ \left( \tfrac{1}{2}/(1 - \pi(x)) \right)^2 - \tfrac{1}{2}/(1 - \pi(x)) \right]$$

$$= \frac{1}{4 \, \pi(x) \left[ 1 - \pi(x) \right]} - 1 \; \geq \; 0,$$

with equality $(\Delta(x) = 0)$ *iff* $\pi(x) = \tfrac{1}{2}$ (observational data already perfectly balanced at that $x$).

Therefore
$$\text{Var}_{\text{IPW}} - \text{Var}_{\text{IWDD}} \; = \; \mathbb{E}_{p_{\text{data}}(x)} \left[ \Delta(x) \, \Sigma(x) \right] \; \succeq \; 0,$$

because each matrix $\Sigma(x)$ is positive–semidefinite and $\Delta(x) \geq 0$.

$\square$

## B  RISK DOMINANCE OF DISTILLATION

We prove that the IWDD generator $G_\theta$ minimizes the target (RCT) test MSE within $\mathcal{G}$ when the distillation objective is Fisher-divergence (score-matching) under Gaussian corruption, and hence achieves no larger test risk than any competitor in $\mathcal{G}$, including the pretrain $f_\phi$.

Let $\mathcal{G} \subseteq \{ g : \mathcal{X} \times \{0, 1\} \to \mathbb{R}^d \}$ be the hypothesis class. For $g \in \mathcal{G}$ define the training risks

$$R_{\text{train}}(g) = \mathbb{E}_{(X,Z,Y) \sim p_{\text{train}}}\big[\|Y - g(X, Z)\|_2^2\big], \quad R_{\text{train}}^{(w)}(g) = \mathbb{E}_{(X,Z,Y) \sim p_{\text{train}}}\big[w(X, Z)\|Y - g(X, Z)\|_2^2\big],$$

with importance weight $w(x, z) = \frac{p_{\text{RCT}}(z)\, p(x)}{p_{\text{train}}(x, z)} > 0$. The target (RCT) test risk is

$$R_{\text{test}}(g) = \mathbb{E}_{(X,Z,Y) \sim p_{\text{test}}}\big[\|Y - g(X, Z)\|_2^2\big], \qquad p_{\text{test}}(x, z) = p(x)\, p_{\text{RCT}}(z).$$

We define the joint test distribution by $p_{\text{test}}(x, z, y) = p(y \mid x, z)\, p(x)\, p_{\text{RCT}}(z)$, i.e., the conditional $p(y \mid x, z)$ is invariant across train and test. Let $f_\phi \in \arg\min_{g \in \mathcal{G}} R_{\text{train}}(g)$ (pretrain) and $G_\theta \in \arg\min_{g \in \mathcal{G}} R_{\text{train}}^{(w)}(g)$ (IWDD generator). Write $q_{g,\sigma}(\cdot \mid x, z)$ for the *Gaussian location* model $\mathcal{N}(g(x, z), \sigma^2 I)$ and let $p_\sigma(\cdot \mid x, z)$ denote the corruption of $p(\cdot \mid x, z)$ by $Y_t = Y + \sigma\varepsilon$, $\varepsilon \sim \mathcal{N}(0, I)$, independent of $(X, Z, Y)$. Fix $\sigma > 0$.

**Assumptions.** (i) Overlap: $0 < w(X, Z) \leq W < \infty$ $p_{\text{train}}$-a.s.; (ii) $\mathbb{E}[\|Y\|_2^2] < \infty$; (iii) $\mathcal{G}$ is nonempty and the map $g \mapsto R_{\text{test}}(g)$ attains its minimum over $\mathcal{G}$.

We first state several auxiliary lemmas that will be used in the proof.

**Lemma 2** (Change of measure / importance weighting). *For any integrable $\varphi(X, Z, Y)$,*

$$\mathbb{E}_{p_{\text{train}}}\big[w(X, Z)\, \varphi(X, Z, Y)\big] = \mathbb{E}_{p_{\text{test}}}\big[\varphi(X, Z, Y)\big].$$

*Proof.* By definition $w(x, z) = \frac{p_{\text{test}}(x, z)}{p_{\text{train}}(x, z)}$, so $\int \varphi\, w\, dp_{\text{train}} = \int \varphi\, dp_{\text{test}}$. $\qquad\square$

**Lemma 3** (Tweedie identity). *For $Y_t = Y + \sigma\varepsilon$ we have*

$$\nabla_{y_t} \log p_\sigma(y_t \mid x, z) = \sigma^{-2}\big(\mathbb{E}[Y \mid y_t, x, z] - y_t\big).$$

**Lemma 4** (Fisher divergence equals test MSE up to a constant). *Define the (unweighted) conditional Fisher objective*

$$\mathcal{J}_{\text{test}}(g) := \mathbb{E}_{(X,Z) \sim p_{\text{test}}} \mathbb{E}_{Y|X,Z} \mathbb{E}_{Y_t|Y}\Big[\|\nabla_{y_t} \log q_{g,\sigma}(Y_t \mid X, Z) - \nabla_{y_t} \log p_\sigma(Y_t \mid X, Z)\|_2^2\Big].$$

*Then*

$$\mathcal{J}_{\text{test}}(g) = \sigma^{-4}\, R_{\text{test}}(g)\, -\, C_\sigma, \qquad C_\sigma := \sigma^{-4}\, \mathbb{E}_{p_{\text{test}}} \mathbb{E}_{Y_t|X,Z}\big[\|Y - \mathbb{E}[Y \mid Y_t, X, Z]\|_2^2\big],$$

*where $C_\sigma \geq 0$ is independent of $g$.*

*Proof.* For the Gaussian location model $q_{g,\sigma}(\cdot \mid x, z) = \mathcal{N}(g(x, z), \sigma^2 I)$, the model score is

$$\nabla_{y_t} \log q_{g,\sigma}(y_t \mid x, z) = \sigma^{-2}\big(g(x, z) - y_t\big).$$

Let $Y_t = Y + \sigma\varepsilon$, $\varepsilon \sim \mathcal{N}(0, I)$ independent of $(X, Z, Y)$, and denote $p_\sigma(y_t \mid x, z) = \int \mathcal{N}(y_t; y, \sigma^2 I)\, p(y \mid x, z)\, dy$. By Lemma 3 (Tweedie identity), for each $(x, z)$, true score is

$$\nabla_{y_t} \log p_\sigma(y_t \mid x, z) = \sigma^{-2}\big(\mathbb{E}[Y \mid y_t, x, z] - y_t\big).$$

*(Justification of differentiating under the integral.)* The Gaussian kernel is $C^\infty$ and $\nabla_{y_t} \mathcal{N}(y_t; y, \sigma^2 I) = \sigma^{-2}(y - y_t)\, \mathcal{N}(y_t; y, \sigma^2 I)$. Since $p(\cdot \mid x, z) \in L^1$ and the Gaussian tails dominate $(1 + \|y\|)$, dominated convergence (or standard mollifier properties) allows swapping $\nabla_{y_t}$ with the $y$-integral.

Subtracting the model score and true score, the score difference is

$$\nabla_{y_t} \log q_{g,\sigma}(Y_t \mid X, Z) - \nabla_{y_t} \log p_\sigma(Y_t \mid X, Z) = \sigma^{-2}\Big(g(X, Z) - \mathbb{E}[Y \mid Y_t, X, Z]\Big).$$

Therefore,

$$\|\nabla_{y_t} \log q_{g,\sigma}(Y_t \mid X, Z) - \nabla_{y_t} \log p_\sigma(Y_t \mid X, Z)\|_2^2 = \sigma^{-4} \left\| g(X, Z) - m(X, Z, Y_t) \right\|_2^2,$$

where we abbreviate $m(X, Z, Y_t) := \mathbb{E}[Y \mid Y_t, X, Z]$.

We here prove the $L^2$-projection (Pythagorean) identity. Write $Y = m + U$ with $U := Y - m$. Then $\mathbb{E}[U \mid Y_t, X, Z] = 0$ by definition of $m$. For any $g = g(X, Z)$, expand

$$\|Y - g\|_2^2 = \|U + (m - g)\|_2^2 = \|U\|_2^2 + \|m - g\|_2^2 + 2\, U^\top (m - g).$$

Taking expectations and using measurability of $m - g$ w.r.t. $\sigma(Y_t, X, Z)$,

$$\mathbb{E}\big[U^\top (m - g)\big] = \mathbb{E}\Big[\mathbb{E}\big[U^\top (m - g) \mid Y_t, X, Z\big]\Big] = \mathbb{E}\Big[(m - g)^\top \underbrace{\mathbb{E}[U \mid Y_t, X, Z]}_{=\,0}\Big] = 0.$$

Thus $\mathbb{E}\|Y - g\|_2^2 = \mathbb{E}\|U\|_2^2 + \mathbb{E}\|m - g\|_2^2$, i.e. $\mathbb{E}\|m - g\|_2^2 = \mathbb{E}\|Y - g\|_2^2 - \mathbb{E}\|U\|_2^2$.

Therefore $\mathbb{E}\|m - g\|_2^2 = \mathbb{E}\|Y - g\|_2^2 - \mathbb{E}\|Y - m\|_2^2$. All quantities are finite by the assumptions $\mathbb{E}\|Y\|_2^2 < \infty$ and $\mathbb{E}\|g(X, Z)\|_2^2 < \infty$ (the latter holds for $g \in \mathcal{G}$ since $R_{\text{test}}(g) < \infty$).

Using this identity, we therefore have

$$\mathbb{E}\big[\|g(X, Z) - m(X, Z, Y_t)\|_2^2\big] = \mathbb{E}\big[\|Y - g(X, Z)\|_2^2\big] - \mathbb{E}\big[\|Y - m(X, Z, Y_t)\|_2^2\big]. \tag{11}$$

Thus,

$$\mathcal{J}_{\text{test}}(g) = \mathbb{E}_{(X,Z)\sim p_{\text{test}}} \mathbb{E}_{Y\mid X,Z} \mathbb{E}_{Y_t\mid Y} \Big[\|\nabla_{y_t} \log q_{g,\sigma}(Y_t \mid X, Z) - \nabla_{y_t} \log p_\sigma(Y_t \mid X, Z)\|_2^2\Big]$$

$$= \mathbb{E}_{(X,Z)\sim p_{\text{test}}} \mathbb{E}_{Y\mid X,Z} \mathbb{E}_{Y_t\mid Y} \Big[\sigma^{-4} \left\| g(X, Z) - m(X, Z, Y_t) \right\|_2^2\Big]$$

$$= \sigma^{-4} \mathbb{E}\Big[\|g(X, Z) - m(X, Z, Y_t)\|_2^2\Big]$$

$$= \sigma^{-4}\big(\mathbb{E}\|Y - g(X, Z)\|_2^2 - \mathbb{E}\|Y - m(X, Z, Y_t)\|_2^2\big)$$

$$= \sigma^{-4} R_{\text{test}}(g) - C_\sigma,$$

where

$$C_\sigma := \sigma^{-4} \mathbb{E}_{p_{\text{test}}} \mathbb{E}_{Y_t\mid Y} \big[\|Y - \mathbb{E}[Y \mid Y_t, X, Z]\|_2^2\big] \ \geq 0$$

depends only on the true distribution and $\sigma$, not on $g$. $\qquad\square$

**Proposition 1** (Weighted Fisher objective equals test MSE up to a constant)**.** *Let*

$$\mathcal{J}_w(g) := \mathbb{E}_{(X,Z)\sim p_{\text{train}}}\big[w(X, Z)\, \mathbb{E}_{Y\mid X,Z} \mathbb{E}_{Y_t\mid Y} \big[\|\nabla_{y_t} \log q_{g,\sigma}(Y_t \mid X, Z) - \nabla_{y_t} \log p_\sigma(Y_t \mid X, Z)\|_2^2\big]\big].$$

*Then $\mathcal{J}_w(g) = \sigma^{-4} R_{\text{test}}(g) - C_\sigma$.*

*Proof.* Apply Lemma 2 to the outer expectation in Lemma 4. $\qquad\square$

[Risk dominance of IWDD] Let $g_{\text{G}} \in \arg\min_{g\in\mathcal{G}} \mathcal{J}_w(g)$. Then, by Proposition 1,

$$R_{\text{test}}(g_{\text{G}}) \ \leq \ R_{\text{test}}(g) \quad \text{for all } g \in \mathcal{G},$$

and in particular $R_{\text{test}}(g_{\text{G}}) \leq R_{\text{test}}(f_\phi)$.

*Proof.* By Proposition 1 (and Lemma 4),

$$\mathcal{J}_w(g) \ = \ \sigma^{-4} R_{\text{test}}(g) \ - \ C_\sigma \quad \text{for all } g \in \mathcal{G},$$

where $\sigma^{-4} > 0$ and $C_\sigma$ does not depend on $g$. Hence for any $g_1, g_2 \in \mathcal{G}$,

$$R_{\text{test}}(g_1) \ \leq \ R_{\text{test}}(g_2) \iff \sigma^{-4} R_{\text{test}}(g_1) - C_\sigma \ \leq \ \sigma^{-4} R_{\text{test}}(g_2) - C_\sigma \iff \mathcal{J}_w(g_1) \ \leq \ \mathcal{J}_w(g_2).$$

That is, the positive affine transformation $r \mapsto \sigma^{-4} r - C_\sigma$ preserves the ordering. Therefore the argmin sets of $R_{\text{test}}$ and $\mathcal{J}_w$ over $\mathcal{G}$ coincide:

$$\arg\min_{g\in\mathcal{G}} \mathcal{J}_w(g) \ = \ \arg\min_{g\in\mathcal{G}} R_{\text{test}}(g).$$

By assumption (iii), $R_{\text{test}}$ attains its minimum over $\mathcal{G}$, so any minimizer $g_{\text{G}}$ of $\mathcal{J}_w$ is a minimizer of $R_{\text{test}}$. Consequently,

$$R_{\text{test}}(g_{\text{G}}) \ \leq \ R_{\text{test}}(g) \quad \text{for all } g \in \mathcal{G},$$

and in particular $R_{\text{test}}(g_{\text{G}}) \ \leq \ R_{\text{test}}(f_\phi)$ for the (possibly different) pretrain minimizer $f_\phi \in \arg\min_{g\in\mathcal{G}} R_{\text{train}}(g)$. $\qquad\square$

**Strict inequality.** We record two equivalent and commonly used sufficient conditions.

**Lemma 5** (Strict dominance when the pretrain is not a test minimizer). *If $f_\phi \notin \arg\min_{g \in \mathcal{G}} R_{\text{test}}(g)$, then there exists $\bar{g} \in \mathcal{G}$ with $R_{\text{test}}(\bar{g}) < R_{\text{test}}(f_\phi)$, and thus*

$$R_{\text{test}}(g_{\text{G}}) \leq R_{\text{test}}(\bar{g}) < R_{\text{test}}(f_\phi).$$

*Hence the inequality in Theorem B is strict.*

*Proof.* By definition of "not a minimizer", there exists $\bar{g}$ with strictly smaller test risk than $f_\phi$. Since $g_{\text{G}}$ minimizes $R_{\text{test}}$, $R_{\text{test}}(g_{\text{G}}) \leq R_{\text{test}}(\bar{g})$. Combining the two inequalities yields the strict chain. $\square$

**Lemma 6** (Directional derivative / quadratic descent criterion). *Let $h := \bar{g} - f_\phi$ and suppose*

$$\mathbb{E}_{p_{\text{test}}}\big[(Y - f_\phi(X,Z))^\top h(X,Z)\big] > 0.$$

*Assume $Y$, $f_\phi(X,Z)$, and $h(X,Z)$ are square-integrable so that all expectations are finite. Then there exists $t^\star \in (0, \infty)$ such that*

$$R_{\text{test}}(f_\phi + t^\star h) < R_{\text{test}}(f_\phi).$$

*Consequently, if $f_\phi + t^\star h \in \mathcal{G}$ (for example, if $\mathcal{G}$ is convex and $f_\phi, \bar{g} \in \mathcal{G}$), then $R_{\text{test}}(g_{\text{G}}) < R_{\text{test}}(f_\phi)$.*

*Proof.* Consider the quadratic expansion, valid since $R_{\text{test}}(g) = \mathbb{E}\|Y - g(X,Z)\|^2$:

$$R_{\text{test}}(f_\phi + th) = R_{\text{test}}(f_\phi) - 2t\,A + t^2\,B,$$

where

$$A := \mathbb{E}_{p_{\text{test}}}\big[(Y - f_\phi(X,Z))^\top h(X,Z)\big], \quad B := \mathbb{E}_{p_{\text{test}}}\big[\|h(X,Z)\|^2\big] \geq 0.$$

If $A > 0$, then necessarily $B > 0$: indeed, if $B = 0$ then $h = 0$ almost surely, forcing $A = 0$, a contradiction. Hence $B > 0$.

For any $t \in (0, 2A/B)$ we have

$$R_{\text{test}}(f_\phi + th) - R_{\text{test}}(f_\phi) = t\,(-2A + tB) < 0.$$

Thus there exists $t^\star > 0$ with strictly smaller risk than $f_\phi$; in fact the optimal quadratic step is $t^\star = A/B$ with decrease $-A^2/B < 0$.

Finally, since $g_{\text{G}}$ minimizes $R_{\text{test}}$ over $\mathcal{G}$, if $f_\phi + t^\star h \in \mathcal{G}$ then

$$R_{\text{test}}(g_{\text{G}}) \leq R_{\text{test}}(f_\phi + t^\star h) < R_{\text{test}}(f_\phi).$$

$\square$

*Remarks.*

(i) Lemma 6 is the standard directional-derivative test: the Gateaux derivative at $f_\phi$ in direction $h$ equals $-2A$, so $A > 0$ guarantees a valid descent step.

(ii) The argument does not require uniqueness of the $R_{\text{test}}$ minimizer; if it is unique, then any Fisher minimizer $g_{\text{G}}$ must coincide with it.

(iii) When $p_{\text{test}} \neq p_{\text{train}}$ (i.e., $w(\cdot)$ non-constant), the minimizers of $R_{\text{train}}$ and $R_{\text{test}}$ generally differ, so unless $f_\phi$ already minimizes $R_{\text{test}}$, strict dominance applies.

## C  TOY DATA GENERATION SETUP

We design a synthetic data generating mechanism to evaluate causal estimation methods under covariate shift, with a particular focus on the generalization of treatment effect estimation from observational (confounded) settings to randomized (unconfounded) settings. In our toy setup, covariate shift arises from changes in the treatment assignment mechanism: the covariate distribution $p(x)$ is fixed across domains, but the conditional treatment distribution $p(z \mid x)$ differs between training and test environments. This setup is conceptually related to the toy example designed for studying classical covariate shift scenarios in supervised learning (Sugiyama et al., 2007), where the input distribution $p(x)$ changes for training and testing while the conditional outcome model $p(y \mid x)$ remains invariant.

Let $x \in \mathbb{R}$ denote a scalar covariate sampled identically across both training and test environments from a standard normal distribution, *i.e.*, $x \sim \mathcal{N}(0, 1)$. The treatment assignment mechanism, however, differs between the training and test datasets.

In the training data, treatment is assigned deterministically based on the covariate:

$$z = \mathbf{1}\{x < -1\},$$

where $\mathbf{1}\{\cdot\}$ is the indicator function. This deterministic rule induces strong confounding, as treatment assignment is a function of $x$. This setup mimics observational studies in which patients with lower health scores or greater severity are more likely to receive treatment.

In contrast, the test data simulates a randomized controlled trial (RCT) setting, where treatment assignment is independent of covariates:

$$z \sim \text{Bernoulli}(0.5).$$

This shift in treatment mechanism introduces a covariate distribution mismatch between $p_{\text{train}}(z \mid x)$ and $p_{\text{test}}(z \mid x)$, despite $p(x)$ remaining unchanged.

The outcome variable $y$ is generated from a nonlinear structural equation that depends on both the covariate $x$ and the treatment $z$:

$$y = \sin(2x) + z \cdot \exp(x) + \epsilon, \quad \epsilon \sim \mathcal{N}(0, 0.01).$$

The function $\sin(2x)$ introduces bounded nonlinear variation in the baseline outcome, while the multiplicative term $z \cdot \exp(x)$ captures heterogeneous treatment effects that increase exponentially with the covariate $x$. The additive noise term $\epsilon$ introduces mild stochasticity, simulating natural outcome variability.

This data generating process encapsulates several key challenges in real-world causal inference: (i) covariate-dependent confounding in observational data, (ii) covariate shift between observational and experimental domains, (iii) heterogeneous treatment effects, and (iv) nonlinear outcome surfaces. As such, it can be used for evaluating the robustness and generalization ability of causal inference methods under distributional mismatch.

# D   ABLATION STUDY AND PARAMETER SETTINGS

We conduct an ablation study to investigate the impact of the hyperparameter $\alpha$ on the performance of IWDD. Tables 3 summarize results across a range of $\alpha \in [0.3, 1.2]$ for the IHDP and ACIC 2018 datasets, respectively.

On the IHDP benchmark, model performance is relatively stable for $\alpha \in [0.3, 0.7]$, with lowest RMSE and Wasserstein distances achieved around $\alpha = 0.7$, while larger values such as $\alpha = 1.2$ lead to slightly degraded metrics. PEHE remains consistent across all $\alpha$ values.

Similarly, on ACIC 2018, RMSE are minimized when $\alpha \in [0.6, 1.2]$, particularly peaking at $\alpha = 0.7$ and $\alpha = 1.2$, whereas values below 0.5 or exactly at 1.0 show inferior results. Interestingly, PEHE shows very little sensitivity to $\alpha$, remaining around 0.300 across the board.

Based on these observations, we use $\alpha = 0.7$ for overall robustness, though task-specific tuning between $\alpha = 0.5$ and $\alpha = 1.2$ may further optimize performance.

Table 3: Average out-of-sample evaluation results for the IWDD algorithm under different values of $\alpha$ on IHDP (10 datasets) and ACIC 2018 (12 datasets)

| $\alpha$ | IHDP | | | ACIC 2018 | | |
|---|---|---|---|---|---|---|
| | $\text{RMSE}_{y_0}$ | $\text{RMSE}_{y_1}$ | PEHE | $\text{RMSE}_{y_0}$ | $\text{RMSE}_{y_1}$ | PEHE |
| 0.3 | 1.197 | 1.115 | 1.645 | 0.947 | 1.204 | 0.300 |
| 0.4 | 1.126 | 1.042 | 1.644 | 0.923 | 1.152 | 0.301 |
| 0.5 | 1.162 | 1.065 | 1.643 | 0.945 | 1.185 | 0.302 |
| 0.6 | 1.399 | 1.305 | 1.643 | 0.895 | 1.019 | 0.300 |
| 0.7 | 1.056 | 0.958 | 1.644 | 0.874 | 1.019 | 0.299 |
| 1.0 | 1.193 | 1.090 | 1.643 | 1.082 | 1.209 | 0.300 |
| 1.2 | 1.236 | 1.154 | 1.642 | 0.889 | 0.986 | 0.301 |

# E IMPLEMENTATION DETAILS

## E.1 IWDD

We implemented IWDD in PyTorch and conducted experiments on an NVIDIA RTX A5000 GPU. Below, we report the default settings of our model, though some hyperparameters may require minor tuning depending on the dataset.

**Pretraining diffusion model** The EDM loss is:

$$\mathcal{L}_\phi = \mathbb{E}_{\sigma,y,n} \left[ \lambda(\sigma) \| D_\phi(y_t; \sigma, x, z) - y \|_2^2 \right],$$

where $D_\phi(y; \sigma, x, z) = c_{\text{skip}}(\sigma) \, y_t \; + \; c_{\text{out}}(\sigma) \, f_\phi\big(c_{\text{in}}(\sigma) \, y_t, \, c_{\text{noise}}(\sigma), x, z\big)$.

We incorporate the diffusion scheduling approach from EDM (Karras et al., 2022). We adopt the same network architecture and preconditioning scheme as EDM, including loss weighting $\lambda(\sigma) = (\sigma^2 + \sigma_{\text{data}}^2)/(\sigma \cdot \sigma_{\text{data}})^2$, input scaling $c_{\text{in}}(\sigma) = 1/\sqrt{\sigma^2 + \sigma_{\text{data}}^2}$, output scaling $c_{\text{out}}(\sigma) = \sigma \cdot \sigma_{\text{data}}/\sqrt{\sigma^2 + \sigma_{\text{data}}^2}$, skip scaling $c_{\text{skip}}(\sigma) = \sigma_{\text{data}}^2/(\sigma^2 + \sigma_{\text{data}}^2)$, and noise conditioning $c_{\text{noise}}(\sigma) = \ln(\sigma)$. All relevant hyperparameters, including $\sigma_{\text{min}}$, $\sigma_{\text{max}}$, $\rho$, and $\mathcal{P}_{\text{mean}}$, are adopted from the EDM default configuration.

For data preprocessing, we followed the approach used in DiffPO (`https://github.com/yccm/DiffPO`). To guide the model, we used three causal masks as inputs: observational ($m_o$), target ($m_t$), and conditional ($m_c$) masks. $m_o$ indicates available observational data, $m_c$ marks conditioning variables $x$ and $a$, and $m_t$ marks observed outcomes $y$. The loss is computed only where $m_t = 1$.

**Distillation** The distillation phase used hyperparameters consistent with the SiD implementation (Zhou et al., 2024). We have the loss functions for generator Eq. 6 and fake score network Eq. 7: The weighting function $w(t)$ is defined as:

$$w(t) = C/\|y_g - f_\phi(y_t, t)\|_{1,\text{sg}},$$

where $y_t = y_g + \sigma_t \epsilon_t$, $C$ is the normalization constant in the structured setting, and $\| \cdot \|_{1,\text{sg}}$ denotes the stop-gradient L1 norm. The same function is used for $\gamma(t)$, following (Karras et al., 2022).

The same hyperparameters are used as in the SiD implementation (Zhou et al., 2024) are used. At each step, we sample $t \sim \text{Unif}[0, t_{\text{max}}/1000]$ with $t_{\text{max}} \in [0, 1000]$ and define the noise level using the $\rho$-parameterized EDM schedule:

$$\sigma_t = \left( \sigma_{\text{max}}^{1/\rho} + (1 - t) \left( \sigma_{\text{min}}^{1/\rho} - \sigma_{\text{max}}^{1/\rho} \right) \right)^\rho,$$

where $\sigma_{\text{min}} = 0.002$, $\sigma_{\text{max}} = 80$, and $\rho = 7.0$.

In the generation procedure $y_g = G_\theta(\sigma_{\text{init}}, x, z, \epsilon), \quad \epsilon \sim \mathcal{N}(0, I)$, $\sigma_{\text{init}}$ is set to 2.5 and remains fixed throughout distillation and evaluation.

## E.2 BASELINES

We compared IWDD against several baselines implemented using publicly available codebases. DiffPO was included with its official implementation and default hyperparameters (`https://github.com/yccm/DiffPO`). CATENets-based estimators—including S-learner, T-learner, DR-learner, RA-learner, TNet, TARNet, OffsetNet, and FlexTENet—were adopted without modification from the CATENets repository (`https://github.com/AliciaCurth/CATENets/tree/main`). GANITE (Yoon et al., 2018), a generative-adversarial baseline for counterfactual prediction, was implemented using the MLforHealthLab repository (`https://github.com/vanderschaarlab/mlforhealthlabpub/tree/main/alg/ganite`). All baseline models were trained and evaluated using the same data splits, preprocessing pipelines, and evaluation metrics as IWDD.

## F    PERFORMANCE METRICS DEFINITION

We evaluate models on three metrics:

- **Root Mean Squared Error (RMSE)** for potential outcome (PO) prediction:

$$\text{RMSE} = \sqrt{\frac{1}{N} \sum_{i=1}^{N} (\hat{y}_i - y_i)^2},$$

  where $\hat{y}_i$ and $y_i$ are the predicted and true outcomes. Lower RMSE indicates better predictive accuracy.

- **Precision in Estimation of Heterogeneous Effect (PEHE)** for treatment effect estimation:

$$\epsilon_{\text{PEHE}} = \sqrt{\frac{1}{N} \sum_{i=1}^{N} \left(\hat{\tau}(x_i) - \tau(x_i)\right)^2},$$

  where $\tau(x) = \mathbb{E}[Y(1) - Y(0) \mid X = x]$ is the Conditional Average Treatment Effect (CATE). Lower values indicate more accurate estimation.

- **Win rate**, reported as a percentage, measuring how often a method outperforms competing approaches.

## G    EXTENDED DISCUSSION OF BASELINE BEHAVIORS

Although OffsetNet and FlexTENet (Curth & van der Schaar, 2021a) perform well in-sample, they show weaker performance in the out-of-sample setting. This is due to how they are designed. OffsetNet uses a hard reparametrization approach, modeling the treatment effect as an additive offset to $\mathbb{E}[Y(0) \mid X]$, explicitly fitting the heterogeneity between POs. FlexTENet employs a multi-task learning architecture with shared and private subspaces to balance common and outcome-specific patterns. Both methods risk overfitting, limiting their ability to generalize. In contrast, IWDD generalizes well across diverse datasets by IPW-modulated distillation through randomization-based adjustment. This is further illustrated by results on ACIC 2016 (Appendix I.1), where IWDD performs better than OffsetNet and FlexTENet.

More specifically, OffsetNet directly estimates the treatment effect $\tau(x)$ as an additive offset to $\mu_0(x)$, giving the model explicit control over fitting the heterogeneity between potential outcomes. This direct parameterization can make OffsetNet highly effective at tightly fitting the training data, especially when the additive assumption aligns well with the data structure. However, this same mechanism can become fragile when the assumed additive relationship does not hold or when $\mu_1(x)$ is inherently simpler than $\mu_0(x)$, leading to reduced generalization out of sample. FlexTENet, on the other hand, employs a flexible multi-task learning architecture with both shared and private subspaces, which allows the model to automatically allocate capacity between what is common across potential outcomes and what is outcome-specific. This flexibility enables FlexTENet to capture complex patterns and reduce in-sample error by adapting to the training data. However, this adaptability also increases the risk of overfitting, especially when the model allocates too much capacity to private, outcome-specific subspaces, thus hurting out-of-sample generalization.

It is worth noting that DiffPO is highly unstable due to its DDPM-based schedules. Although it performs well on some ACIC 2018 datasets, it fails on all ACIC 2016 and IHDP datasets, and is thus excluded from those results due to lack of comparability[5]. Although propensity weights, which should always exceed 1, are clipped to 0.9 for numerical stability, its DDPM-based schedules remains unstable in some settings. The EDM schedules we adopt, however, contributes to robustness.

---

[5]We use the official GitHub repository of DiffPO to produce the results: `https://github.com/yccm/DiffPO`.

# H Theory Under Overlap Violation

Standard weighting estimators such as IPW or AIPW are ill-defined when the overlap assumption fails, since the inverse propensities diverge or collapse to zero. In such cases causal effects are not identifiable from observational data, and risk-based estimators cannot be constructed. IWDD, however, does not rely on calculated weights. Instead, it directly minimizes the expected divergence to the true conditional outcome distribution under the randomized test policy $q(x, z) = p_{\text{data}}(x)\, p_{\text{rct}}(z)$, which remains well-defined even under deterministic treatment assignment.

Let $p_{\text{data}}(x)$ denote the covariate marginal in the training data and $p_{\text{rct}}(z)$ the randomized test policy. Define the *target law*

$$q(x, z) = p_{\text{data}}(x)\, p_{\text{rct}}(z).$$

Let $p(y \mid x, z)$ be the outcome mechanism, $f_\phi(y \mid x, z)$ the pretrained diffusion model (teacher), and $q_\theta(y \mid x, z)$ the IWDD model. IWDD trains by minimizing the expected divergence to the true conditional distribution under the target law:

$$\mathcal{L}^{\text{IWDD}}(\theta) = \mathbb{E}_{(x,z)\sim q}\Big[D\Big(q_\theta(\cdot \mid x, z) \,\Big\|\, p(\cdot \mid x, z)\Big)\Big],$$

where $D(\cdot\|\cdot)$ denotes the conditional Fisher divergence in our implementation, though the results apply to any divergence that is nonnegative and vanishes if and only if the two conditionals agree almost surely.

**Assumptions.**

- (S0) (**Stable outcome mechanism**) The conditional distribution of outcomes is invariant: $p(y \mid x, z)$ is the same in training and test. Equivalently, only the joint distribution of covariates and treatment $(X, Z)$ shifts between train and test, so while the marginal distribution of $Y$ may change, the conditional mechanism generating $Y$ given $(X, Z)$ remains stable.

- (S1) (**Finite energy**) $\mathcal{L}^{\text{IWDD}}(\theta) < \infty$ for the students considered and $\mathbb{E}_q[D(p_\phi\|p)] < \infty$.

- (S2) (**Optimization accuracy**) The trained student $\hat{\theta}$ satisfies $\mathcal{L}^{\text{IWDD}}(\hat{\theta}) \leq \inf_\theta \mathcal{L}^{\text{IWDD}}(\theta) + \varepsilon$ for some small $\varepsilon \geq 0$.

Let $\mathcal{G} = \{q_\theta(\cdot \mid x, z) : \theta \in \Theta\}$ denote the generator hypothesis class and define the best-in-class (target) divergence

$$\mathcal{L}^\star = \inf_\theta \mathbb{E}_{(x,z)\sim q}\Big[D\Big(q_\theta(\cdot \mid x, z) \,\Big\|\, p(\cdot \mid x, z)\Big)\Big].$$

**Theorem 3** (Strict improvement over the pretrained model on the target law without overlap assumption). *Under (S0)–(S2), if the generator class $\mathcal{G}$ contains a model that is closer to the true conditional distribution under the target law q than the pretrained model, i.e.*

$$\mathcal{L}^\star \leq \mathbb{E}_{(x,z)\sim q}\Big[D\Big(p_\phi(\cdot \mid x, z) \,\Big\|\, p(\cdot \mid x, z)\Big)\Big] - \eta \quad \text{for some } \eta > 0,$$

*then the trained IWDD generator $\hat{\theta}$ achieves a strictly smaller target divergence than the pretrained model:*

$$\mathbb{E}_{(x,z)\sim q}\Big[D\Big(q_{\hat{\theta}}(\cdot \mid x, z) \,\Big\|\, p(\cdot \mid x, z)\Big)\Big] \leq \mathbb{E}_{(x,z)\sim q}\Big[D\Big(p_\phi(\cdot \mid x, z) \,\Big\|\, p(\cdot \mid x, z)\Big)\Big] - (\eta - \varepsilon).$$

*In particular, if $\varepsilon < \eta$ we obtain* strict *improvement on the target law. Crucially, this guarantee holds even without overlap.*

*Proof.* By (S2), $\mathcal{L}^{\text{IWDD}}(\hat{\theta}) \leq \mathcal{L}^\star + \varepsilon$. By the assumed gap $\eta > 0$,

$$\mathcal{L}^\star + \varepsilon \leq (\mathbb{E}_q[D(p_\phi \,\|\, p)] - \eta) + \varepsilon = \mathbb{E}_q[D(p_\phi \,\|\, p)] - (\eta - \varepsilon).$$

Since the left-hand side equals $\mathbb{E}_q[D(q_{\hat{\theta}}\|p)]$ by definition of $\mathcal{L}^{\text{IWDD}}$, the claim follows. $\qquad\square$

**When is $\eta > 0$ reasonable?** The gap $\eta > 0$ arises whenever the class $\mathcal{G}$ admits a model that better fits the *target law* than the teacher. This is natural in our setting because: (i) the pretrained model $f_\phi$ is fit under the observational policy and may be misaligned with the randomized target distribution; and (ii) IWDD explicitly trains under the target law $q(x, z) = p_{\text{data}}(x)p_{\text{rct}}(z)$, so the best-in-class student can reduce target divergence. In contrast to weighting-based estimators, which become undefined without overlap, the IWDD objective remains well-posed, making this improvement guarantee robust even under deterministic treatment assignment.

# I  ADDITIONAL EXPERIMENTS RESULTS

## I.1  ACIC 2016

We present full potential outcome prediction results for ten selected datasets from the 77 ACIC 2016 datasets (Table 4) and provide detailed treatment effect estimation results for three representative datasets (Table 5).

## I.2  ACIC 2018

We present the complete RMSE results for each of the 12 ACIC 2018 datasets in Table 6, with in-sample and out-of-sample RMSE reported for the POs $Y(0)$ and $Y(1)$. The full PEHE results for each dataset are provided in Table 7.

Table 4: RMSE for POs $Y(0)$ and $Y(1)$ (in-sample and out-of-sample) across 10 ACIC 2016 datasets. The best result across all methods is highlighted in **bold**.

| | Dataset 1 | | | | Dataset 2 | | | |
| --- | --- | --- | --- | --- | --- | --- | --- | --- |
| | $RMSE_{0,in}$ | $RMSE_{0,out}$ | $RMSE_{1,in}$ | $RMSE_{1,out}$ | $RMSE_{0,in}$ | $RMSE_{0,out}$ | $RMSE_{1,in}$ | $RMSE_{1,out}$ |
| TNet | 1.3069 | 1.4667 | 1.7358 | 1.8146 | 2.552 | 2.684 | 4.717 | 4.786 |
| TNet_reg | 1.3157 | 1.4827 | 1.5979 | 1.6294 | 2.442 | 2.448 | 3.295 | 3.210 |
| TARNet | 1.2697 | 1.4360 | 1.4534 | 1.5463 | 2.343 | 2.320 | 2.941 | 2.872 |
| TARNet_reg | 1.2481 | 1.4148 | 1.3863 | 1.5026 | 2.251 | 2.213 | 2.607 | 2.578 |
| OffsetNet | 1.2469 | 1.4302 | 1.4231 | 1.5514 | 2.229 | 2.198 | 2.409 | 2.306 |
| FlexTENet | 1.1800 | 1.3420 | 1.2402 | 1.3453 | 2.140 | 1.995 | 2.522 | 2.388 |
| FlexTENet_noortho | 1.3290 | 1.5117 | 1.5809 | 1.6845 | 2.377 | 2.415 | 3.004 | 2.932 |
| Pretrain | 0.954 | 0.981 | 1.007 | 1.300 | 0.855 | 1.028 | 1.302 | 1.414 |
| IWDD | **0.952** | **0.938** | **0.997** | **1.153** | **0.846** | **1.023** | **1.171** | **1.274** |

| | Dataset 3 | | | | Dataset 4 | | | |
| --- | --- | --- | --- | --- | --- | --- | --- | --- |
| | $RMSE_{0,in}$ | $RMSE_{0,out}$ | $RMSE_{1,in}$ | $RMSE_{1,out}$ | $RMSE_{0,in}$ | $RMSE_{0,out}$ | $RMSE_{1,in}$ | $RMSE_{1,out}$ |
| TNet | 0.8678 | 1.0245 | 1.5526 | 1.6606 | 2.3082 | 2.2645 | 3.1790 | 3.8997 |
| TNet_reg | 0.8422 | 1.0140 | 1.2557 | 1.3984 | 2.2534 | 2.2559 | 2.3215 | 3.1703 |
| TARNet | 0.8283 | 0.9890 | 1.2168 | 1.3560 | 2.1608 | 2.1516 | 2.1011 | 2.9622 |
| TARNet_reg | 0.7996 | 0.9720 | 1.0432 | 1.2079 | 2.0858 | 2.0906 | 2.0797 | 2.9229 |
| OffsetNet | 0.7968 | 0.9852 | **0.9391** | **1.1258** | 2.1010 | 2.1452 | 2.1386 | 3.0270 |
| FlexTENet | **0.7776** | 0.9690 | 1.1019 | 1.2620 | 2.0541 | 2.0146 | 2.2698 | 3.2725 |
| FlexTENet_noortho | 0.8699 | 1.0731 | 1.2849 | 1.4463 | 2.1818 | 2.2343 | 2.3238 | 3.2638 |
| Pretrain | 0.963 | **0.868** | 1.753 | 1.428 | 0.918 | 1.001 | 1.090 | 0.979 |
| IWDD | 0.954 | 0.871 | 1.735 | 1.250 | **0.909** | **0.986** | **1.132** | **0.890** |

| | Dataset 5 | | | | Dataset 6 | | | |
| --- | --- | --- | --- | --- | --- | --- | --- | --- |
| | $RMSE_{0,in}$ | $RMSE_{0,out}$ | $RMSE_{1,in}$ | $RMSE_{1,out}$ | $RMSE_{0,in}$ | $RMSE_{0,out}$ | $RMSE_{1,in}$ | $RMSE_{1,out}$ |
| TNet | 1.834 | 2.081 | 2.747 | 2.824 | 1.212 | 1.481 | 2.564 | 2.676 |
| TNet_reg | 1.941 | 2.211 | 2.296 | 2.420 | 1.245 | 1.434 | 1.931 | 2.039 |
| TARNet | 1.953 | 2.229 | 2.122 | 2.278 | 1.218 | 1.416 | 1.850 | 1.905 |
| TARNet_reg | 1.921 | 2.208 | 1.977 | 2.187 | 1.159 | 1.392 | 1.651 | 1.770 |
| OffsetNet | 1.915 | 2.260 | 2.035 | 2.238 | 1.139 | 1.392 | 1.545 | 1.719 |
| FlexTENet | 1.747 | 2.021 | 1.797 | 1.961 | 1.094 | 1.325 | 1.623 | 1.725 |
| FlexTENet_noortho | 2.014 | 2.329 | 2.134 | 2.363 | 1.252 | 1.465 | 1.963 | 2.030 |
| Pretrain | 0.907 | **1.235** | 0.991 | 1.058 | 0.964 | **0.993** | 1.194 | **1.290** |
| IWDD | **0.875** | 1.305 | 1.007 | **1.031** | 0.962 | 1.027 | **1.189** | 1.384 |

| | Dataset 7 | | | | Dataset 26 | | | |
| --- | --- | --- | --- | --- | --- | --- | --- | --- |
| | $RMSE_{0,in}$ | $RMSE_{0,out}$ | $RMSE_{1,in}$ | $RMSE_{1,out}$ | $RMSE_{0,in}$ | $RMSE_{0,out}$ | $RMSE_{1,in}$ | $RMSE_{1,out}$ |
| T-learner | 2.328 | 1.971 | 3.233 | 2.827 | 2.395 | 2.854 | 2.540 | 3.695 |
| S-learner | 2.301 | 1.956 | 3.429 | 3.006 | 2.408 | 2.883 | 2.523 | 3.716 |
| TNet | 2.075 | 2.534 | 3.548 | 4.173 | 2.279 | 3.985 | 3.479 | 3.932 |
| TNet_reg | 1.496 | 3.231 | 2.853 | 4.074 | 2.081 | 3.807 | 3.052 | 3.644 |
| TARNet | 1.486 | 3.204 | 2.754 | 3.983 | 1.899 | 3.858 | 2.576 | 3.308 |
| TARNet_reg | 1.459 | 3.172 | 2.633 | 3.874 | 1.875 | 3.840 | 2.430 | 3.248 |
| OffsetNet | 1.596 | 3.311 | 2.671 | 3.853 | 1.897 | 3.706 | 2.422 | 3.111 |
| FlexTENet | 1.424 | 3.133 | 2.533 | 3.740 | 1.778 | 3.340 | 2.402 | 3.062 |
| FlexTENet_noortho | 1.538 | 3.243 | 2.796 | 4.039 | 2.004 | 3.841 | 2.812 | 3.624 |
| Pretrain | 0.923 | 0.971 | 1.246 | 1.230 | 1.038 | 1.315 | 1.452 | 1.877 |
| IWDD | **0.920** | **0.956** | **1.231** | **1.135** | **0.968** | **1.100** | **1.352** | **1.591** |

| | Dataset 9 | | | | Dataset 10 | | | |
| --- | --- | --- | --- | --- | --- | --- | --- | --- |
| | $RMSE_{0,in}$ | $RMSE_{0,out}$ | $RMSE_{1,in}$ | $RMSE_{1,out}$ | $RMSE_{0,in}$ | $RMSE_{0,out}$ | $RMSE_{1,in}$ | $RMSE_{1,out}$ |
| TNet | 2.835 | 3.048 | 3.852 | 3.861 | 1.724 | 2.037 | 3.708 | 3.858 |
| TNet_reg | 2.843 | 3.089 | 3.576 | 3.790 | 1.735 | 1.959 | 2.573 | 2.874 |
| TARNet | 2.762 | 3.057 | 3.305 | 3.565 | 1.644 | 1.970 | 2.252 | 2.508 |
| TARNet_reg | 2.717 | 3.056 | 3.168 | 3.471 | 1.579 | 1.880 | 2.051 | 2.304 |
| OffsetNet | 2.685 | 3.035 | 3.100 | 3.501 | 1.613 | 1.973 | 2.187 | 2.502 |
| FlexTENet | 2.508 | 2.821 | 3.037 | 3.324 | 1.505 | 1.659 | 2.058 | 2.332 |
| FlexTENet_noortho | 2.748 | 3.081 | 3.289 | 3.568 | 1.688 | 2.019 | 2.424 | 2.735 |
| Pretrain | 1.095 | **1.066** | 0.904 | 1.006 | 0.950 | 1.068 | 1.437 | 1.407 |
| IWDD | **1.085** | 1.198 | 0.906 | **0.962** | **0.931** | **1.062** | **1.392** | **1.396** |

Table 5: PEHE (in-sample and out-of-sample) on ACIC 2016-2, 2016-7, and 2016-26

| Algorithm | 2016-2 | | 2016-7 | | 2016-26 | |
|---|---|---|---|---|---|---|
| | $PEHE_{in}$ | $PEHE_{out}$ | $PEHE_{in}$ | $PEHE_{out}$ | $PEHE_{in}$ | $PEHE_{out}$ |
| Causal Forest (CF) | **0.322** | **0.320** | 3.832 | 2.969 | 2.613 | 3.234 |
| T-learner | 1.071 | 1.054 | 3.605 | 2.790 | 2.493 | 3.015 |
| S-learner | 0.906 | 0.873 | 3.902 | 3.075 | 2.721 | 3.357 |
| TNet | 4.114 | 4.315 | 4.180 | 4.377 | 3.534 | 4.219 |
| TNet_reg | 2.528 | 2.560 | 2.961 | 5.256 | 3.218 | 3.984 |
| TARNet | 2.157 | 2.164 | 2.899 | 5.230 | 2.695 | 3.485 |
| TARNet_reg | 1.605 | 1.577 | 2.740 | 5.104 | 2.506 | 3.268 |
| OffsetNet | 1.454 | 1.464 | 2.865 | 5.221 | 2.600 | 3.248 |
| FlexTENet | 1.466 | 1.450 | 2.660 | 5.026 | 2.544 | 3.569 |
| FlexTENet_noortho | 2.205 | 2.199 | 2.931 | 5.258 | 2.930 | 3.833 |
| DRNet | 2.402 | 2.552 | 3.611 | 5.890 | 3.215 | 3.916 |
| DRNet_TAR | 1.594 | 1.599 | 3.154 | 5.427 | 2.825 | 3.526 |
| Pretrain | 0.849 | 0.851 | **1.128** | **1.034** | **1.377** | **1.562** |
| **IWDD** | 0.852 | 0.851 | 1.130 | 1.036 | 1.387 | **1.562** |

Table 6: RMSE for potential outcomes $Y(0)$ and $Y(1)$ (in-sample and out-of-sample) across 12 ACIC 2018 datasets. Each 4-column block corresponds to one dataset, labeled by its ID. The best result across all methods is highlighted in **bold**, and the best result among the diffusion-based approaches (DiffPO, Pretrain, IWDD) is additionally marked with a ⋆.

|  | e36aca1030264e638452ea4053cbb42c | | | | d4ae3280e4e24ca395533e429726fafc | | | |
|---|---|---|---|---|---|---|---|---|
|  | $RMSE_{0,in}$ | $RMSE_{0,out}$ | $RMSE_{1,in}$ | $RMSE_{1,out}$ | $RMSE_{0,in}$ | $RMSE_{0,out}$ | $RMSE_{1,in}$ | $RMSE_{1,out}$ |
| T-learner | 273.048 | 353.999 | 273.043 | 353.783 | 2.138 | 2.165 | 2.067 | 2.167 |
| S-learner | 264.114 | 353.674 | 264.136 | 353.746 | 2.241 | 2.280 | 2.202 | 2.293 |
| TNet | 0.872 | 1.119 | 0.877 | 1.118 | 0.520 | 0.545 | 0.528 | 0.565 |
| TARNet | 0.874 | 1.113 | 0.887 | 1.130 | 0.532 | 0.550 | 0.515 | 0.573 |
| OffsetNet | 0.836 | 1.163 | 0.852 | 1.204 | **0.472** | 0.614 | 0.554 | 0.681 |
| FlexTENet | **0.818** | 1.155 | **0.836** | 1.185 | 0.475 | 0.565 | **0.486** | 0.607 |
| DiffPO | 1.025 | 1.043 | 1.025 | 1.045 | 0.509 | 0.454 | 0.509⋆ | **0.454**⋆ |
| Pretrain | 1.019 | 1.037 | 1.021 | 1.040 | 0.517 | 0.459 | 1.697 | 1.697 |
| IWDD | 1.012⋆ | **1.028**⋆ | 1.014⋆ | **1.030**⋆ | 0.503⋆ | **0.447**⋆ | 1.786 | 1.782 |

|  | d1546da12d8e4daf8fe6771e2187954d | | | | ae51149d38ce42609e00bf5701e4fe88 | | | |
|---|---|---|---|---|---|---|---|---|
|  | $RMSE_{0,in}$ | $RMSE_{0,out}$ | $RMSE_{1,in}$ | $RMSE_{1,out}$ | $RMSE_{0,in}$ | $RMSE_{0,out}$ | $RMSE_{1,in}$ | $RMSE_{1,out}$ |
| T-learner | 48.249 | 2763.573 | 50.768 | 2829.238 | 17.130 | 18.984 | 17.288 | 18.481 |
| S-learner | 47.393 | 2763.576 | 48.612 | 2829.269 | 16.731 | 18.683 | 16.802 | 18.335 |
| TNet | 0.092 | 1.282 | 0.179 | 1.322 | 1.017 | 1.124 | 1.003 | 1.073 |
| TARNet | 0.058 | 1.278 | 0.109 | 1.311 | 1.038 | 1.153 | 0.993 | 1.069 |
| OffsetNet | **0.057** | 1.279 | **0.058** | 1.309 | 0.976 | 1.306 | 0.890 | 1.200 |
| FlexTENet | 0.884 | 1.055 | 1.076 | 1.203 | 1.029 | 1.296 | 0.927 | 1.117 |
| DiffPO | 1.105 | 0.027 | 1.131 | 0.026 | **0.958**⋆ | **1.050**⋆ | **0.957**⋆ | **1.051**⋆ |
| Pretrain | 1.105 | 0.028 | 1.131 | 0.029 | 0.996 | 1.067 | 0.980 | 1.052 |
| IWDD | 1.105 | **0.026**⋆ | 1.131 | **0.026** ⋆ | 0.997 | 1.067 | 0.979 | 1.051 ⋆ |

|  | ac6e494cbc254dc599be26a2a17f229c | | | | 9333a461d3944d089ef60cdf3b88fd40 | | | |
|---|---|---|---|---|---|---|---|---|
|  | $RMSE_{0,in}$ | $RMSE_{0,out}$ | $RMSE_{1,in}$ | $RMSE_{y_1,out}$ | $RMSE_{0,in}$ | $RMSE_{0,out}$ | $RMSE_{1,in}$ | $RMSE_{1,out}$ |
| T-learner | 15.834 | 16.559 | 15.882 | 16.962 | 7.754 | 8.281 | 9.244 | 9.427 |
| S-learner | 15.503 | 16.432 | 15.391 | 16.621 | 7.585 | 8.161 | 8.689 | 8.943 |
| TNet | 0.993 | 1.050 | 1.066 | 1.147 | 1.008 | 1.073 | 1.184 | 1.214 |
| TARNet | 0.996 | 1.059 | 1.073 | 1.154 | 1.000 | 1.087 | 1.195 | 1.244 |
| OffsetNet | **0.903** | 1.144 | 1.022 | 1.313 | **0.910** | 1.191 | 1.073 | 1.333 |
| FlexTENet | 0.927 | 1.096 | 1.033 | 1.271 | 0.949 | 1.149 | 1.119 | 1.265 |
| DiffPO | 1.004⋆ | 1.020 | **1.004**⋆ | 1.021 | 0.995⋆ | **1.056**⋆ | **0.999**⋆ | **1.058**⋆ |
| Pretrain | 1.147 | 1.126 | 1.239 | 1.213 | 1.070 | 1.113 | 1.215 | 1.252 |
| IWDD | 1.010 | **0.970**⋆ | 1.045 | **0.994**⋆ | 1.016 | 1.101 | 1.125 | 1.232 |

|  | 8ff38d337ec842dab1b8c01076e24816 | | | | 74420a1794304013bb7a5a8f61994d71 | | | |
|---|---|---|---|---|---|---|---|---|
|  | $RMSE_{0,in}$ | $RMSE_{0,out}$ | $RMSE_{1,in}$ | $RMSE_{1,out}$ | $RMSE_{0,in}$ | $RMSE_{0,out}$ | $RMSE_{1,in}$ | $RMSE_{1,out}$ |
| T-learner | 21.940 | 21.629 | 21.018 | 21.543 | 182.881 | 187.618 | 186.001 | 187.741 |
| S-learner | 21.199 | 21.207 | 20.632 | 21.232 | 179.791 | 187.670 | 181.017 | 187.627 |
| TNet | 1.079 | 1.098 | 1.042 | 1.090 | 1.008 | 1.047 | 1.026 | 1.050 |
| TARNet | 1.070 | 1.077 | 1.042 | 1.096 | 1.014 | 1.037 | 1.039 | 1.045 |
| OffsetNet | **0.970** | 1.218 | **0.951** | 1.244 | **0.909** | 1.168 | **0.916** | 1.195 |
| FlexTENet | 1.014 | 1.106 | 0.984 | 1.127 | 0.923 | 1.141 | 0.944 | 1.124 |
| DiffPO | 2344.775 | 2302.331 | 4969.306 | 4861.299 | 1.050 | 1.007 | 1.051 | 1.004 |
| Pretrain | 1.063 | 1.063 | 1.068 | 1.068 | 1.045 | 0.990 | 1.045 | 0.990 |
| IWDD | 1.046⋆ | **1.050**⋆ | 1.049⋆ | **1.052**⋆ | 1.023 ⋆ | **0.970**⋆ | 1.024⋆ | **0.970**⋆ |

|  | 110f6dc8583c456ea0dd242d5d598497 | | | | 3ebc51612e034ff99e8632a228dae430 | | | |
|---|---|---|---|---|---|---|---|---|
|  | $RMSE_{0,in}$ | $RMSE_{0,out}$ | $RMSE_{1,in}$ | $RMSE_{y_1,out}$ | $RMSE_{0,in}$ | $RMSE_{0,out}$ | $RMSE_{1,in}$ | $RMSE_{1,out}$ |
| T-learner | 0.404 | 0.416 | **0.716** | **0.731** | 33.291 | 34.316 | 36.608 | 36.009 |
| S-learner | **0.395** | **0.415** | 0.727 | 0.747 | 33.082 | 34.230 | 34.997 | 35.051 |
| TNet | 0.723 | 0.754 | 1.271 | 1.313 | 1.006 | 1.029 | 1.126 | 1.147 |
| TARNet | 0.727 | 0.756 | 1.264 | 1.302 | 1.017 | 1.034 | 1.119 | 1.102 |
| OffsetNet | 0.669 | 0.883 | 1.186 | 1.384 | **0.920** | 1.145 | 1.016 | 1.210 |
| FlexTENet | 0.685 | 0.810 | 1.213 | 1.343 | 0.963 | 1.096 | 1.059 | 1.151 |
| DiffPO | 1.272 | 1.277 | 1.874 | 1.885 | 128.388 | 94.869 | 133.157 | 99.067 |
| Pretrain | 0.760 | 0.745 | 1.505 | 1.481 | 0.998 | 1.020 | 0.997 | 1.019 |
| IWDD | 0.753⋆ | 0.717⋆ | 1.486⋆ | 1.402⋆ | 0.995⋆ | **1.018**⋆ | 0.996⋆ | 1.018 ⋆ |

|  | 5a147c7e542a4ea5b22da127b654666b | | | | 5ad181455e954bcba44743e1f2d7824e | | | |
|---|---|---|---|---|---|---|---|---|
|  | $RMSE_{0,in}$ | $RMSE_{0,out}$ | $RMSE_{1,in}$ | $RMSE_{y_1,out}$ | $RMSE_{0,in}$ | $RMSE_{0,out}$ | $RMSE_{1,in}$ | $RMSE_{1,out}$ |
| T-learner | 129.082 | 142.823 | 136.431 | 144.381 | 59.179 | 62.521 | 69.447 | 68.557 |
| S-learner | 127.209 | 142.641 | 129.352 | 142.609 | 58.596 | 62.059 | 65.345 | 65.455 |
| TNet | 0.957 | 1.056 | 1.133 | 1.195 | 1.005 | **1.066** | 1.132 | 1.125 |
| TARNet | 0.943 | 1.057 | 1.124 | 1.187 | 0.995 | 1.068 | 1.141 | 1.137 |
| OffsetNet | **0.870** | 1.128 | 1.104 | 1.356 | **0.941** | 1.136 | **1.046** | 1.181 |
| FlexTENet | 0.892 | 1.088 | 1.082 | 1.232 | 0.958 | 1.138 | 1.084 | 1.166 |
| DiffPO | 3204.518 | 3267.525 | 6589.601 | 6733.434 | 1.119 | 1.118 | 1.380 | 1.141 |
| Pretrain | 0.993 | 1.058 | 0.990 | 1.056 | 1.157 | 1.117 | 1.148 | 1.123 |
| IWDD | 0.987⋆ | **1.053**⋆ | **0.989**⋆ | **1.053**⋆ | 1.109⋆ | 1.113⋆ | 1.110⋆ | **1.113**⋆ |

Table 7: PEHE (in-sample and out-of-sample) across 12 ACIC 2018 datasets. Each 2-column block corresponds to one dataset, labeled by its ID (truncated). The best performance across all methods is marked in **bold**, and the best among diffusion-based methods is marked with $\star$.

|  | e36aca10... | | d4ae3280... | | d1546da1... | |
|---|---|---|---|---|---|---|
|  | $\text{PEHE}_{in}$ | $\text{PEHE}_{out}$ | $\text{PEHE}_{in}$ | $\text{PEHE}_{out}$ | $\text{PEHE}_{in}$ | $\text{PEHE}_{out}$ |
| Causal Forest | 15.577 | 12.722 | 0.690 | 0.684 | 4.354 | 65.782 |
| T-learner | 25.286 | 15.863 | 0.696 | 0.679 | 6.140 | 65.813 |
| S-learner | 5.704 | 4.951 | 1.609 | 1.611 | 1.966 | 65.705 |
| TNet | 0.370 | 0.343 | 0.331 | 0.318 | 0.165 | 0.153 |
| TARNet | 0.416 | 0.410 | 0.385 | 0.412 | 0.125 | 0.135 |
| OffsetNet | 0.437 | 0.437 | 0.361 | 0.349 | 0.003 | 0.030 |
| FlexTENet | 0.397 | 0.397 | 0.313 | 0.311 | 0.692 | 0.687 |
| DRNet | 0.406 | 0.400 | 0.325 | 0.356 | 0.177 | 0.196 |
| DiffPO | 0.032 | 0.053 | **0.025**$^\star$ | **0.017**$^\star$ | 0.031 | 0.009 |
| Pretrain | **0.011**$^\star$ | **0.011**$^\star$ | 1.744 | 1.742 | 0.026 | **0.001**$^\star$ |
| IWDD | **0.011**$^\star$ | **0.011**$^\star$ | 1.741 | 1.741 | **0.001**$^\star$ | **0.001**$^\star$ |

|  | ae51149... | | ac6e494... | | 9333a461... | |
|---|---|---|---|---|---|---|
|  | $\text{PEHE}_{in}$ | $\text{PEHE}_{out}$ | $\text{PEHE}_{in}$ | $\text{PEHE}_{out}$ | $\text{PEHE}_{in}$ | $\text{PEHE}_{out}$ |
| Causal Forest | 9.740 | 9.686 | 7.499 | 7.505 | 7.480 | 7.481 |
| T-learner | 9.722 | 9.602 | 7.597 | 7.545 | 7.531 | 7.520 |
| S-learner | 7.823 | 7.744 | 5.188 | 5.167 | 6.115 | 6.099 |
| TNet | 0.673 | 0.657 | 0.686 | 0.679 | 0.995 | 0.983 |
| TARNet | 0.744 | 0.737 | 0.723 | 0.739 | 1.030 | 1.033 |
| OffsetNet | 0.798 | 0.779 | 0.739 | 0.750 | 0.948 | 0.950 |
| FlexTENet | 0.841 | 0.834 | 0.717 | 0.728 | 1.000 | 1.003 |
| DRNet | 0.721 | 0.719 | 0.648 | 0.653 | 0.981 | 0.967 |
| DiffPO | **0.016**$^\star$ | **0.026**$^\star$ | **0.022**$^\star$ | **0.019**$^\star$ | **0.038**$^\star$ | **0.021**$^\star$ |
| Pretrain | 0.238 | 0.230 | 0.183 | 0.181 | 0.333 | 0.310 |
| IWDD | 0.229 | 0.229 | 0.195 | 0.181 | 0.320 | 0.322 |

|  | 8ff38d... | | 74420a... | | 110f6dc... | |
|---|---|---|---|---|---|---|
|  | $\text{PEHE}_{in}$ | $\text{PEHE}_{out}$ | $\text{PEHE}_{in}$ | $\text{PEHE}_{out}$ | $\text{PEHE}_{in}$ | $\text{PEHE}_{out}$ |
| Causal Forest | 12.302 | 12.298 | 13.113 | 11.653 | 0.523 | **0.525** |
| T-learner | 12.390 | 12.318 | 19.321 | 13.813 | **0.521** | **0.525** |
| S-learner | 9.436 | 9.382 | 8.398 | 6.172 | 0.560 | 0.563 |
| TNet | 0.717 | 0.718 | 0.404 | 0.382 | 0.594 | 0.588 |
| TARNet | 0.734 | 0.743 | 0.448 | 0.456 | 0.617 | 0.613 |
| OffsetNet | 0.701 | 0.714 | 0.465 | 0.475 | 0.563 | 0.556 |
| FlexTENet | 0.667 | 0.676 | 0.448 | 0.449 | 0.612 | 0.603 |
| DRNet | 0.714 | 0.712 | 0.426 | 0.414 | 0.684 | 0.680 |
| DiffPO | 3590.180 | 3585.164 | 0.017 | 0.054 | 0.828$^\star$ | 0.830$^\star$ |
| Pretrain | 0.014 | 0.014 | **0.002**$^\star$ | **0.002**$^\star$ | 1.032 | 1.023 |
| IWDD | **0.009**$^\star$ | **0.009**$^\star$ | **0.002**$^\star$ | **0.002**$^\star$ | 1.044 | 1.032 |

|  | 3ebc516... | | 5a147c... | | 5ad181... | |
|---|---|---|---|---|---|---|
|  | $\text{PEHE}_{in}$ | $\text{PEHE}_{out}$ | $\text{PEHE}_{in}$ | $\text{PEHE}_{out}$ | $\text{PEHE}_{in}$ | $\text{PEHE}_{out}$ |
| Causal Forest | 14.542 | 14.531 | 33.209 | 32.968 | 42.400 | 42.450 |
| T-learner | 14.957 | 14.743 | 35.354 | 33.761 | 42.580 | 42.509 |
| S-learner | 10.452 | 10.354 | 14.072 | 13.650 | 32.033 | 32.030 |
| TNet | 0.707 | 0.667 | 0.706 | 0.667 | 0.731 | 0.729 |
| TARNet | 0.699 | 0.672 | 0.699 | 0.672 | 0.790 | 0.809 |
| OffsetNet | 0.767 | 0.756 | 0.767 | 0.756 | 0.721 | 0.721 |
| FlexTENet | 0.676 | 0.633 | 0.676 | 0.633 | 0.785 | 0.795 |
| DRNet | 0.716 | 0.698 | 0.716 | 0.698 | 0.773 | 0.783 |
| DiffPO | 5.037 | 4.198 | 3402.876 | 3479.084 | 0.932 | 0.232 |
| Pretrain | 0.106 | 0.084 | 0.103 | 0.069 | **0.027**$^\star$ | **0.0001**$^\star$ |
| IWDD | **0.052**$^\star$ | **0.052**$^\star$ | **0.034**$^\star$ | **0.034**$^\star$ | 0.057 | **0.0001**$^\star$ |

## I.3 IHDP

Tables 9 and 8 summarize the results on the IHDP dataset. IWDD achieved the best out-of-sample PEHE performance. However, since the IHDP dataset contains only 747 samples, several baselines outperformed IWDD on RMSE. This is likely because IWDD has a large number of parameters to optimize, and IHDP is a relatively small dataset.

Table 8: IHDP1: RMSE

| Algorithm | RMSE | | | |
|---|---|---|---|---|
| | $Y(0)$ (in) | $Y(0)$ (out) | $Y(1)$ (in) | $Y(1)$ (out) |
| T-learner | 2.862 | 3.287 | **0.355** | **0.357** |
| S-learner | 2.976 | 3.403 | 1.393 | 1.334 |
| TNet | 0.716 | 1.415 | 0.995 | 0.982 |
| TNet_reg | 0.825 | 1.644 | 1.052 | 1.000 |
| TARNet | 0.804 | 1.639 | 1.028 | 0.957 |
| TARNet_reg | 0.816 | 1.650 | 0.959 | 0.890 |
| OffsetNet | 0.913 | 1.704 | 1.667 | 1.376 |
| FlexTENet | 0.757 | 1.550 | 0.985 | 0.865 |
| FlexTENet_noortho | 0.794 | 1.631 | 1.109 | 1.003 |
| Pretrain | **0.527** | **0.533** | 1.608 | 1.630 |
| IWDD | 0.545 | 0.597 | 1.789 | 1.902 |

Table 9: PEHE (in-sample and out-of-sample) on IHDP1

| Algorithm | PEHE$_{in}$ | PEHE$_{out}$ |
|---|---|---|
| Causal Forest (CF) | 4.072 | 4.265 |
| T-learner | 2.690 | 3.102 |
| S-learner | 3.695 | 3.966 |
| TNet | 1.255 | 1.886 |
| TNet_reg | 1.365 | 2.070 |
| TARNet | 1.336 | 2.028 |
| TARNet_reg | 1.290 | 2.010 |
| OffsetNet | 1.771 | 2.320 |
| FlexTENet | **1.142** | 1.786 |
| FlexTENet_noortho | 1.375 | 2.064 |
| DRNet | 1.276 | 1.933 |
| DRNet_TAR | 1.277 | 1.741 |
| GANITE | 1.923 | 2.433 |
| Pretrain | 1.681 | 1.658 |
| IWDD | 1.698 | **1.655** |

