# OpenReview forum: "A Generative Framework for Causal Estimation via Importance-Weighted Diffusion Distillation"
_ICLR.cc/2026/Conference — Submitted to ICLR 2026_

### Official Review · Reviewer_BX17 · 2025-10-20

**Soundness:** 3
**Presentation:** 2
**Contribution:** 2
**Rating:** 4
**Confidence:** 5

**Summary:**

The authors proposed Importance-Weighted Diffusion Distillation (IWDD), a two stage framework for estimating causal effect in the binary treatment setting. In the first stage, a conditional diffusion model is trained to learn P(Y|Z,X) and also serves as the teacher network. In the second stage, a distillation process is conducted by introducing a fake score network together with the teacher network to guide the training of a one-step generator. Although it is not an end-to-end framework, the method IWDD overcomes some key limitations in previous works, such as DiffPO, where a propensity score model is not required by considering the imbalance through sampling in an implicit way. The theoretical analysis and empirical results look promising.

**Strengths:**

1.	The two-stage method is pretty straight forward and written in a clear way.
2.	Incorporating the randomized control and considering sample weight in an implicit way by a simple sampling process seems a novel contribution.
3.	The empirical evaluation is pretty compelling by comparing to quite a few solid baselines across several different simulation datasets.
4.	The theoretical results are technically sound, especially the compared of different variance estimator and also the empirical risk.

**Weaknesses:**

1.	The current work only handle binary treatment setting, which is quite limited since complex treatments, such as continuous or categorical are pretty common in real datasets.
2.	Although unconfoundedness is a standard assumption but still quite strong. How unmeasured confounders can impact the the performance of IWDD is not discussed. E.g., will the proposed method be more robust to unmeasured confounder compared to other baselines?
3.	The first stage of training a conditional diffusion model may be computational extensive. This could also be a potential barrier for real applications.
4.	The proposed method did not consider the high-dimensionality case of X. If simply concatenating X and Z for conditioning, the high-dimensional X may dominant the information since Z is just one-dimensional.
5.	Some important baselines that also use generative AI are missed. E.g., GANITE, CausalEGM based on GAN, and CEVAE based on VAE. Especially considering CausalEGM is a quite recent method and seems to outperform existing baselines, it should be compared and benchmarked.

**Questions:**

1.	Could the authors provide the performance drop when introducing unmeasured confounder for different methods? The effect of the unmeasured confounder can be adjusted to different levels
2.	The author should also include the missed baselines, such as GANITE, CausalEGM, and CEVAE for a more comprehensive comparison.
3.	The scalability of the proposed method IWDD should be rigorously tested by varying the sample size.
4.	The author stated that Fisher divergence is better than KL divengence, any intuition for this empirical conclusion?
5.	How the method is sensitive to the dimension of X?
6.	The author claims that one-step generator largely improves efficiency. Then the running time and computational cost at different stages should be benchmarked.
7.	Why causalforest is only shown in Table 2 not Table 1? The performance of causalforest is much worse than other baselines is dubious as it is not consistent from the results in other papers.

---

> ### Author Response · Authors · 2025-11-20
>
> We thank the reviewer for acknowledging our strengths. We respond to weaknesses and questions as follows:
>
> **Weaknesses:**
> [W1]: “IWDD only handles binary treatment setting.”
>
> We follow the standard ACIC/IHDP benchmarks and baselines (OffsetNet, FlexTENet, DRNet, TARNet, DiffPO), all defined for binary treatment. Evaluation metrics such as PEHE and RMSE are also standard in this setting. The IWDD formulation itself does not rely on binary treatment; extending to continuous or multi-valued treatments only requires adapting the conditioning distribution and evaluation setup. This is mentioned in Sec.5: Discussion about the limitations.
>
> [W2]& [Q1]. “How unmeasured confounders can impact the the performance of IWDD is not discussed.”
>
> Our paper adopts the same unconfoundedness assumption as the baselines and benchmarks. Studying performance under simulated unmeasured confounding is a promising direction for future work. This is mentioned in Sec.5: Discussion about the limitations.
>
> [W3] & [Q6]. Running time and computational cost at different stage.
>
> Our method consists of two stages: (1) pretraining a diffusion model and (2) distilling it into a one-step generator. The EDM-based diffusion model is trained for 2000 epochs with a residual MLP-style U-Net backbone (embedding dimension 128, four residual blocks). The distillation phase reuses the pretrained model and jointly trains a one-step generator and a “fake” diffusion model using randomized conditioning and noise-corrupted targets. This stage also runs for approximately 2000 epochs. The total training time remains practical—approximately 3 to 5 minutes on an NVIDIA RTX A5000 GPU. In terms of computational cost, we estimate that IWDD requires approximately 9 TFLOPs to train. (One TFLOP = one trillion floating point operations.)
>
> Compared to other baselines like DiffPO and CATENets, IWDD avoids training a separate propensity score network, which typically adds 500–1000 training iterations and 1–3 minutes of CPU time per dataset. DiffPO trains one DDPM-based diffusion model and an additional propensity score network, the overall training cost for DiffPO is ~9-12 TFLOPs. Moreover, inference with IWDD is substantially more efficient. The distilled generator performs prediction in a single forward pass (1 NFE), compared to 100–1000 denoising steps (100–1000 NFEs) required by DiffPO. This results in a significant speedup at inference time, making IWDD especially suitable for high-throughput or real-time applications.
>
> Compared to CATENets-style methods (e.g., S-learner, T-learner, TARNet, OffsetNet), our method incurs higher training cost due to diffusion and distillation. These models are typically trained with small MLPs (3–5 layers, batch size 100) and converge within 500 epochs, often with early stopping. However, they also require training a separate propensity network, similar in cost to that used by DiffPO. In total, CATENets are cheaper to train (e.g., ~5–8 minutes on CPU), but they underperform in predictive accuracy.
>
> Method	| Training Cost (FLOPs) |	Inference Time
> ---|-----|------
> IWDD	| ~9 TFLOPs |	1 NFE
> DiffPO	| ~9-12 TFLOPs |	100 NFE
> CATENets	| ~1-2 TFLOPs|	-
>
> We will include the training cost and inference time of IWDD and comparisons to other baselines in the section of implementation details in the revised version.
>
> [W4] & [Q5]
>
> > the high-dimensional X may dominant the information since Z is just one-dimensional.
>
> The concern that concatenating a high-dimensional covariate vector X and a 1-dimensional treatment Z may cause X to dominate does not apply to our setting. IWDD is not trained on the joint observational distribution p(x,z); instead, it uses marginal sampling to draw $(x,z) \sim p(x)p(z)$. Under this randomized product distribution, X carries no information about Z, so the model cannot absorb treatment effects into covariates, regardless of dimensionality. Moreover, the distillation process explicitly conditions the score network on Z at every diffusion step, and the semantics of the diffusion targets differ for z=0 and z=1. Thus, Z acts as a structural switching variable, and its influence cannot be dominated by the dimensionality of X. Empirically, IWDD cleanly separates the potential outcome surfaces even in high-dimensional settings. In practice, one often add the projected rerepsentation of $X$ and $Z$ in the same embedding space, further addressing the concerns the dimensionality of X overpowering Z.
>
> [W5] & [Q2] other generative baselines.
>
> To maintain comparability with existing studies, we follow the established evaluation setups that consistently use DRNet, FlexTENet, OffsetNet, TARNet, DiffPO, etc. We will include results for other generative baselines such as CausalEGM and CEVAE in the revision.

---

> > ### Author Response · Authors · 2025-11-20
> >
> > **Questions:**
> > [Q1], [Q2], [Q5] and [Q6] are answered above.
> >
> > [Q3]
> > > The scalability of the proposed method IWDD should be rigorously tested by varying the sample size.
> >
> > ACIC 2018 contains large datasets (n = 10,000 with 177 covariates), which serves as a practical scalability test. IWDD trains stably on all ACIC 2018 datasets under the same hyperparameter grid. We agree scalability is important and will add discussion on this point in the revision.
> >
> > [Q4]
> > > Intuition of why fisher divergence is better than KL divengence.
> >
> > In theory, the optimal solutions for KL divergence and Fisher divergence are the same, both achieved when  $q_\theta(y\mid x, z) = p(y\mid x, z) $. However, empirically we observe that Fisher divergence provides better performance. This aligns with findings in image distillation tasks, where score-based SiD consistently outperforms the KL-based DMD (Zhou et al., 2024) in standard benchmark datasets, such as CIFAR10 and ImageNet.
> >
> > Mingyuan Zhou, Huangjie Zheng, Zhendong Wang, Mingzhang Yin, and Hai Huang. Score identity distillation: Exponentially fast distillation of pretrained diffusion models for one-step generation. In International Conference on Machine Learning, 2024.
> >
> > [Q7]
> > > Why causalforest is only shown in Table 2 not Table 1? The performance of causalforest is much worse than other baselines is dubious.
> >
> > Causal Forest directly estimates conditional average treatment effects rather than potential outcomes. Therefore, it does not produce Y(0) and Y(1), and cannot be evaluated under our metrics for PO estimation used in Table 1. We follow the official implementation and the same evaluation protocol as existing studies. We also note that Causal Forest performs reasonably well in ACIC 2016 (see Table 5 in the appendix).

---

### Official Review · Reviewer_uGJs · 2025-10-29

**Soundness:** 2
**Presentation:** 2
**Contribution:** 1
**Rating:** 2
**Confidence:** 4

**Summary:**

This paper proposes a new framework for causal effect estimation with diffusion models. For that, the authors develop Importance-Weighted Diffusion Distillation (IWDD), which first pretrains diffusion models and then performs balancing of the covariate distributions of the treated and untreated groups by importance sampling, as an alternative to IPW loss adjustment. The authors then perform experiments on different datasets and benchmark their method against related baselines.

**Strengths:**

- Adapting diffusion models for causal effect estimation is an interesting research idea, allowing for going beyond pure point estimation of CATEs but also allowing for estimating the conditional potential outcomes distribution, which then can be used to estimate different causal quantities.
- Large parts of the paper are well written, and the flow of the paper is easy to follow.
- The authors provide code for the reproducibility of their results.

**Weaknesses:**

**The contribution/novelty and practicality of the method seem rather limited.**
- The idea of using diffusion models for potential outcomes estimation while tackling distribution shift/treatment selection bias is not new and has already been explored e.g. by DiffPO. Thus, the major contribution of this paper is avoiding using the IPW term and instead performing importance weighting via sampling. The applied adaptations and performed experiments are also limited and could be improved clearly (see points below).
- One major motivation of their applied weighing via sampling is avoiding IPW weighing the loss term which is stated to be problematic because of (i) the need for estimating the propensity score and (ii) avoiding unstable optimization in case of extreme propensity score weights/overlap violations. However, both points do not seem to be solved better by their proposed framework. For (i), instead of estimating the propensity score (a rather “simple” task, i.e. training a calibrated binary classification model), their method requires a pretrained diffusion model and a fake score network which seems to be a computationally more expensive and also from the estimation side way more complex problem, making the intuition unclear why this should be a better approach. For (ii), as the authors state, IPW can apply simple clipping/trimming with the cost of losing the orthogonal properties of the loss but likely increasing finite sample estimation performance. However, the method of the authors requires also evaluating the pretrained diffusion and fake score network for sampled combinations of (x,z) in regions that were never observed during training, which intuitively should also lead to degraded performance. Here, I think the provided variance analysis is not really useful for showing benefits of their method because it does not take into account the variance for estimating p(y|x,z) and evaluating it in these low overlap regions, which is the main problem of their method here.

**The experimental setup is unclear and raises serious concerns.**
 - Figure 2 seems to have no clear meaning. The paper does conditional potential outcomes distribution estimation, what is the purpose of showing a marginal distribution shift of y between training and test domain?
- Figure 3 is unclear. The true distributions of potential outcomes in orange should a) at least be the same within each column (why are they different?) and b) should be the same for Y(0) for both in-sample and out of sample (and for Y(1) the same). (I guess in sample and out of sample refers to training and test domain.) This is because the plot implies the distribution of _potential_ (not just observed) outcomes are plotted, and there is no distribution shift in x between training and test data. Did the authors plot the distribution of observed outcomes instead? If not, it is unclear to me why the trained model should perform well for predicting Y(1) in-sample but bad for Y(1) out of sample since there is no distribution shift in x. So I guess for in-sample, the authors actually show P(Y|Z=0) and P(Y|Z=1)  following the observational distribution instead.
- Overall and importantly, if the treatment assignment is fully deterministic here (no overlap for any x), both potential outcomes for any x are never identifiable at all so showing results here is actually just stating that for some randomly selected DGP one method is better than the other because of some inductive bias. This is not useful for causal inference method validation.
- The comparison to the baselines seems strange. In Tables 1 and 2, how can DiffPO have notable percentage win rates but incredibly high average error rates? This might be an indicator for some model failure for individual datasets and thus improper hyperparameter tuning. Also the link provided in l.269 trying to show the incorrect implementation of DiffPO does not work properly and does not show what is stated by the authors.

**Questions:**

- In Fig. 2, the paper does conditional potential outcomes distribution estimation, what is the purpose of showing a marginal distribution shift of y between training and test domain?
- In Fig 3., the true distributions of potential outcomes in orange should a) at least be the same within each column, why are they different?
- In Fig. 3, did the authors plot the distribution of observed outcomes instead of potential outcomes for in sample data?
- In Tables 1 and 2, how can DiffPO have notable percentage win rates but incredibly high average error rates?

---

> ### Author Response · Authors · 2025-11-20
>
> We thank the reviewer for the time. The reviewer raises concerns regarding contribution and synthetic experiment setup. We respond below.
>
> ## Responses to Weaknesses
>
> **"Limited contribution/novelty and practicality."**
>
> We respectfully disagree with the reviewer’s assessment.
>
> > “The idea is not new; DiffPO already uses diffusion for causal estimation.”
>
> Our contribution is not the use of diffusion models per se, but the new importance-weighted objective (Lemma 1) that establishes an analytical equivalence between IPW and an unweighted divergence under p(x)p(z).
> This equivalence does not appear in DiffPO or prior diffusion-based causal work and is the foundation for IWDD’s randomized adjustment, reduced gradient variance, and stable training.
> Thus, IWDD is not a modification of DiffPO but a different objective with different theoretical properties. While we fix practical issues in DiffPO—such as its incorrect handling of propensity weighting and its unstable DDPM noise schedule—these are not our main contributions. They simply strengthen the pretrained teacher model. Our core contribution is independent of DiffPO and lies in the principled embedding of IPW into marginal sampling and the resulting distillation framework.
>
> > “The major contribution is avoiding the IPW term; unclear why this is better.” "Propensity estimation is simple; IWDD seems more complex."
>
> Propensity estimation is simple; weighting the loss is not. The computational and statistical challenge in IPW comes from multiplying the loss by inverse propensities, which inflates variance and destabilizes optimization, especially under overlap violations. IWDD avoids this entirely.
>
> > Practicality is limited due to diffusion and a fake score network
>
> The distillation stage produces a one-step generator, replacing the 100–1,000 denoising steps required by multi-step diffusion models. This substantially improves inference efficiency.
>
> To summarize our key contribution and novelty, IWDD introduces a new theoretical objective (Lemma 1) that establishes a rigorous equivalence between IPW and an unweighted divergence under $p(x)p(z)$, enabling a principled randomized adjustment without estimating propensities or weighting the loss. This reparameterization yields provably lower gradient variance and more stable optimization (Theorem 2). Building on this foundation, IWDD further provides a practical one-step generator through distillation, which is proved in theory to have risk dominance compared to pretained model, replacing 100–1,000 diffusion steps with a single forward pass, and delivers a simpler pipeline that avoids a separate propensity model and the instability caused by IPW weighting.
>
> **"The experimental setup is unclear and raises serious concerns."**
>
> > Figure 2 seems to have no clear meaning.
>
> See [Q1].
>
> > Figure 3 is unclear...
>
> See [Q2] & [Q3].
>
> > Overall and importantly, if the treatment assignment is fully deterministic here ...
>
> The motivation of the synthetic experiment with fully deterministic treatment assignment is to demonstrate IWDD's robustness even when overlap assumption is violated. The reviewer’s statement simply restates the well-known fact that deterministic treatment assignment destroys identifiability, something we never dispute and never use this experiment to argue. We support this synthetic experiment with explicit theory for overlap violation in Appendix H.
>
> The remark that "for some randomly selected DGP one method is better than the other because of some inductive bias" overlooks our **main empirical evaluation** in ACIC 2016, ACIC 2018 and IHDP, where IWDD consistently performes better than other baselines.
>
> > The comparison to the baselines seems strange. In Tables 1 and 2, how can DiffPO ... incredibly high average error rates?... improper hyperparameter tuning. Also the link provided in l.269 ... (also in [Q4].)
>
> The fact that DiffPO attains notable win rates yet very large average errors is explained by its per-dataset instability, not hyperparameter tuning. As shown explicitly in Table 6, DiffPO performs competitively on some ACIC 2018 datasets (contributing to its win rate), but on several others, including all ACIC 2016 AND IHDP datasets, leads to complete failures. These catastrophic failures occur under the official implementation and recommended settings. It is not because of the hyperparameter tuning, it is due to the DDPM schedule they adopt. In contrast, IWDD adopts EDM scheduling in pretrained diffusion and performance is robust to hyperparameters. This is explained in details in Appendix G: extended discussion of baseline behaviors.
>
> At the time of writing our submission, the repository contained the version we referenced. An issue was publicly raised on the problematic trimming logic. The issue is no longer visible, and the repository’s commit history has since been rewritten. As a result, the specific link in our submission is now inactive.

---

> > ### Author Response · Authors · 2025-11-20
> >
> > ## Questions:
> > [Q1].
> > > In Fig. 2, ...?
> >
> > Figure 2 is included to illustrate the synthetic setting used in our overlap‐violation experiment. The plot shows how the marginal distribution of y differs between the training (observational) domain and the test domain when the overlap assumption is violated. This helps readers understand the specific scenario we construct where the distribution of outcomes in the test domain differs substantially from the observational data. The figure is therefore used only to visualize the synthetic setup, not to analyze conditional potential outcome distributions.
> >
> > [Q2]
> > > In Fig 3., the true distributions of potential outcomes in orange should a) at least be the same within each column, why are they different?
> >
> > We thank the reviewer for pointing this out. In Figure 3, the orange curves show the observed outcomes in each split, not the full marginal distributions of the potential outcomes. Since the datasets are generated by independent random draws from the same data‐generating mechanism, their empirical distributions can differ slightly, even though the underlying DGP is the same. This is why the orange curves are not pixel‐identical across columns, although they converge to the same true distribution in expectation.
> >
> > [Q3]
> > > In Fig. 3, did the authors plot the distribution of observed outcomes instead of potential outcomes for in sample data?
> >
> > Yes. We plot the distribution of observed outcomes. The purpose of this figure is to illustrate how overlap violation induces a shift in the observational outcome distribution between the training and test domains. We will clarify this in the caption and main text. This clarification does not affect any result or conclusion, as comparisons in real-world observational settings also rely on observed outcomes rather than full potential-outcome distributions.
> >
> > [Q4]
> > > DiffPO performance
> >
> > Answered above.

---

### Official Review · Reviewer_QBWv · 2025-11-01

**Soundness:** 2
**Presentation:** 3
**Contribution:** 1
**Rating:** 2
**Confidence:** 4

**Summary:**

The paper proposes a two-stage framework, which first pretrains a covariate- and treatment-conditional diffusion model on observational data, then incorporates inverse probability weighting (IPW) into the distillation process to adjust for confounding. The method shows theoretical variance reduction and results on empirical datasets. The paper shows good empirical performance on causal learning challenge datasets in ACIC.

**Strengths:**

- The idea to use deep learning techniques such as diffusion and distillation on IPW is innovative, which arguably hasn't been done before
- The paper shows details of how diffusion and distillation are built into the algorithm, and shows the rationale
- The paper shows good results on the increasingly popular ACIC datasets

**Weaknesses:**

- The diffusion/distillation them algorithms are popular algorithms, while it's uncertain how much impact the algorithm has, because it's only applied on the dealing of confounding factors, or IPW
- The paper may be missing key ablation studies, e.g., IPW itself uses one-layer regression, which itself is a shallow ML technique. What if we use an MLP or other deeper models without diffusion or distillation? Ablation to see the contribution to see each technique will be helpful to boost the paper's impact.
- The results could use comparisons with other techniques, SOTA (combining IPW and DL), and against IPW, also in other datasets other than ACIC. ACIC has multiple datasets, but the comparison with other techniques could be seen on other older datasets.

**Questions:**

- Could you add more ablation studies on how diffusion and distillation each contribute to the good results
- Other than diffusion and distillation being popular applications in deep learning, are they the only techniques that could add to performance increase? e.g. p(y | x, z) could be estimated many different ways, with DL, graphical models
- More comparison and analysis on public datasets, with previous methods, e.g., DragonNet, would be helpful.

---

> ### Author Response · Authors · 2025-11-20
>
> We thank the reviewer for acknowledging our strengths. The reviewer’s main concerns focus on the model choice of diffusion distillation, particularly regarding its impact and comparisons. In particular, the reviewer asked for ablations using an MLP instead of diffusion/distillation and for additional comparisons with other techniques and datasets. We respond below.
>
> **"impact unclear; only applied to confounding/IPW" (W1)**
>
> IPW indeed addresses confounding bias, but IWDD is not “only applied to dealing with confounding or IPW.” Our contribution is the reparameterization of the IPW objective (Lemma 1), the variance‐reduction optimization, the theoretical guarantee of risk dominance of distillation, and randomization-based adjustment, all of which form a complete generative causal estimation framework. IWDD is evaluated end-to-end on ACIC 2016, ACIC 2018, and IHDP, demonstrating improved PO prediction and CATE estimation beyond merely correcting confounding.
>
> **"missing ablations of using MLP instead of diffusion." (W2); why diffusion distillation instead of other techniques(Q1 & Q2)**
>
> We do not claim that diffusion and distillation are the only possible techniques. In addition to the theoretical rationale and related literature given in the paper, we summarize here the intuitive reasoning behind IWDD’s design choices.
>
> Motivated by prior work, we begin with diffusion models because they are powerful generative frameworks known to capture complex, high-dimensional data distributions. They have demonstrated strong empirical performance not only in image synthesis but also in predictive tasks such as classification and regression (e.g., CARD by Han et al., 2022). This makes them well-suited to modeling the conditional potential outcome distributions, $p(y \mid x, z)$, in causal inference, hence this is how we train pretrained model.
>
> However, a drawback of diffusion models is high inference cost, for the hundreds of denoising steps. Thus, we incorporate distillation. Our approach follows the line of work in score-based diffusion distillation. We specifically adopt Score Identity Distillation (Zhou et al., 2024), which allows us to efficiently distill from the pretrained model to a one-step generator using Fisher divergence.
>
> A central and novel mechanism in IWDD is importance weighting via marginal sampling during distillation. Instead of learning a separate propensity score model to correct for confounding—as is standard in inverse probability weighting—we embed the reweighting mechanism directly into the distillation training process.
>
> Together, these components, (1) expressive diffusion modeling, (2) efficient one-step distillation, and (3) principled importance-weighted training, complement each other and are jointly motivated by the challenges of robust potential outcome estimation in observational settings.
>
> **[W2], [W3] & [Q3]: More comparison and analysis on public datasets, with previous methods, including other DL techniques.**
>
> Our experiments evaluate on ACIC 2016, ACIC 2018, and IHDP, the standard benchmarks used in recent causal inference work, and include diverse strong baselines such as DRNet, FlexTENet, OffsetNet, TARNet, DiffPO, etc.
>
> References:
>
> Xizewen Han, Huangjie Zheng, and Mingyuan Zhou. CARD: Classification and Regression Diffusion Models. In Thirty-Sixth Conference on Neural Information Processing Systems (NeurIPS 2022).
>
> Mingyuan Zhou, Huangjie Zheng, Zhendong Wang, Mingzhang Yin, and Hai Huang. Score identity distillation: Exponentially fast distillation of pretrained diffusion models for one-step generation. In International Conference on Machine Learning, 2024.

---

### Official Review · Reviewer_3g5t · 2025-11-03

**Soundness:** 1
**Presentation:** 2
**Contribution:** 1
**Rating:** 2
**Confidence:** 4

**Summary:**

The paper introduces a new generative framework for modelling potential outcome distributions, namely, Importance-Weighted Diffusion Distillation (IWDD). The proposed method uses a pre-trained diffusion model to distill its predictions onto a different generative model. Furthermore, during the distillation process, the authors suggested a randomization-based adjustment that serves as a substitute for the inverse probability weighting (IPW). The paper also formulates theoretical guarantees for when the suggested distillation should improve over the baseline diffusion model.  Finally, the authors evaluate their IWDD empirically based on the standard causal ML benchmarks.

**Strengths:**

The proposed idea is original. Also, the paper has a clear structure.

**Weaknesses:**

The method relies on the core idea that we can substitute the IPWs with the randomization-based adjustment (i.e., shuffling the covariates and treatment assignment). Yet, by doing so, we cannot use the observed outcomes from the original dataset (as they originate from P(X, Z, Y) and not from P(X)P(Z)P(Y|X, Z)). The paper also does not clearly explain what sample is being used for the distillation, so I assume the pre-trained diffusion model was used to sample from both of the potential outcomes, P(Y|X, Z). If this is indeed the case, the theoretical insights in Sec. 3.3 are obsolete: There, the authors assume that the target model $G_\Theta$ minimizes the risk wrt. data from the RCT (=P(X)P(Z)P(Y|X, Z)). Yet, in reality (as far as I understood), the ground-truth P(Y|X, Z) is substituted with another diffusion model.  I encourage the authors to clarify these important details.

Therefore, I question the sanity of the proposed method: To the best of my knowledge, there is no direct way to omit the inverse propensity weights (other than trimming/truncation/retargeting of the loss) if we want to use the observational data and want to incorporate the propensity score into the loss.    Furthermore, in my opinion, the paper has limited novelty (e.g., in comparison with Diff-PO [1]), and the main method is simply the implementation of a two-stage covariate-adjusted learner with diffusion models.

I also found several minor mistakes:
- Line 195. $\lambda(\sigma)$ was not defined.
- I double-checked the code from [1], and the propensity scores were trimmed to at least 0.33 in DiffPO (https://github.com/yccm/DiffPO/blob/15a6b675236736a95a3adfdf1711390893b8fd96/src/main_model.py#L150). Also, the link that the authors provided in line 269 is not active anymore.

References:
- [1] Yuchen Ma, Valentyn Melnychuk, Jonas Schweisthal, and Stefan Feuerriegel. DiffPO: A causal diffusion model for learning distributions of potential outcomes. In The Thirty-eighth Annual Conference on Neural Information Processing Systems, 2024. URL https://openreview.net/forum?id=merJ77Jipt.

**Questions:**

- Why were CATE/CAPOs benchmarks chosen if the main task is the estimation of the potential outcomes distributions? I would expect the benchmarks to center around evaluating different distributional distances (e.g., Wasserstein distance). Also, many generative baselines for potential outcomes are missing (e.g., [1, 2]).
- How are the hyperparameters of the main method and other baselines tuned? I haven’t found any details on that in the Appendix and the provided code.

References:
- [1] Yoon, Jinsung, James Jordon, and Mihaela Van Der Schaar. "GANITE: Estimation of individualized treatment effects using generative adversarial nets." International conference on learning representations. 2018.
- [2] Vanderschueren, Toon, Jeroen Berrevoets, and Wouter Verbeke. "NOFLITE: Learning to predict individual treatment effect distributions." Transactions on Machine Learning Research (2023).

---

> ### Author Response · Authors · 2025-11-20
>
> ## Reviewer’s misunderstanding of IPW vs randomization / “sanity of the method”
> **Clarification on the role of IPW, marginal sampling, and the randomization-based adjustment.**
> Our method does **not** omit or replace inverse-propensity weighting (IPW).
> Lemma 1 in the paper shows that the standard IPW objective is  equivalent to the unweighted risk under the product-of-marginals distribution.
> Thus, the randomization-based adjustment is a reparameterization of the IPW objective via marginal sampling, not a substitution or omission of IPW.
>
> **Clarification on the data used in distillation.**
> The reviewer also writes that, after shuffling, “we cannot use the observed outcomes from the original dataset.” In fact, observed outcomes y are only used in the pretraining stage of the teacher diffusion model $f_\phi$ (Section 3.1). This is clearly stated in Algorithm 1. During distillation , we draw $(\tilde x, \tilde z)$ from the randomized product distribution (via shuffling x and sampling $z\sim \text{Bern}(0.5))$, generate $\tilde y_g = G_\theta(\tilde x,\tilde z,\varepsilon)$, and form noisy versions $y_t = \tilde y_g + \sigma_t\varepsilon_t$. The generator and fake-score updates in Equations (6)–(7) involve only $(\tilde x,\tilde z, y_t, \tilde y_g)$ and the teacher scores $f_\phi$; the original observed outcomes do not enter the distillation step.
>
> The reviewer’s doubt about Sec. 3.3 is therefore based on the misunderstanding of the samples we use in the distillation stage.
>
> **"Limited novelty compared to DiffPO"**
>
> While our work indeed improves upon DiffPO in several practical aspects, these are not the main contributions of the paper. Compared to DiffPO, we identify and correct its implementation issue related to propensity weighting, and we resolve DiffPO’s instability by adopting an EDM noise schedule rather than the original DDPM schedule. These improvements strengthen the pretrained teacher model, but they represent only a small part of our contribution.
>
> Our primary contributions go far beyond DiffPO. Independent of DiffPO, we propose a principled embedding of inverse-propensity weighting (IPW) into marginal sampling, establishing a mathematical equivalence that yields both theoretical clarity and empirical gains. Moreover, we introduce a one-step distillation procedure that replaces slow multi-step diffusion sampling, offering substantial speedups in inference while maintaining estimation accuracy.
>
>
> **Claimed minor mistakes:**
> 1. $\lambda(\sigma)$ is defined in Line 1095.
>
> 2. We appreciate the reviewer checking the DiffPO code. At the time of writing our submission, the repository contained the version we referenced. An issue was publicly raised on the problematic trimming logic. The issue is no longer visible, and the repository’s commit history has since been rewritten. As a result, the specific link in our submission is now inactive.
>
> ## Questions:
>
> 1.
> > Why were CATE/CAPOs benchmarks chosen if the main task is the estimation of the potential outcomes distributions?
>
> Our method predicts the full potential outcomes $Y(0)$ and $Y(1)$, and therefore naturally provides CATE as well. For this reason, it is standard to evaluate POs and CATE jointly on these benchmarks since many baseline methods focus on estimating CATE instead of POs.
>
> > I would expect the benchmarks to center around evaluating different distributional distances (e.g., Wasserstein distance).
>
> We've considered evaluation metrics such as wasserstein distance. In this case wasserstein distance is not applicable. Wasserstein distance between full conditional distributions requires access to the true distributions $P(Y\mid X,Z)$, which are not available in these causal benchmarks. Each unit provides only one realization of each potential outcome. As a result, wasserstein distance cannot be meaningfully computed against ground-truth distributions.
>
> > Also, many generative baselines for potential outcomes are missing
>
> We included the major baselines with publicly available, maintained implementations, including DRNet, FlexTENet, OffsetNet, TARNet, DiffPO, etc. We will also incorporate results for additional generative baselines such as GANITE and NOFLITE in the revision.
>
>
> 2.
> > How are the hyperparameters of the main method and other baselines tuned? I haven’t found any details on that in the Appendix and the provided code.
>
> The ablation study for the key hyperparameters of IWDD is provided in **Appendix D**. The parameter controlling the distillation regularization can be tuned via the flag -alpha in the provided code.
> For all baseline methods, we follow the default hyperparameters recommended in their original implementations. This is consistent with standard practice in the literature, as many baselines are optimized for these benchmark datasets.

---

### Meta-Review · Area_Chair_3NmP · 2026-01-07

**Summary:**

This paper proposes a diffusion-based framework for causal estimation from observational data. The key idea is to integrate importance weighting into diffusion score distillation, enabling efficient one-step sampling while correcting for covariate imbalance, while applying randomized adjustment to stabilize the estimation of the propensity score.

However, several concerns remain unresolved, including insufficient clarification of the underlying distributional assumptions, the lack of intuitive explanation for the randomized adjustment and limited discussion on alternative generative backbones.

Overall, despite the research direction is promising, the current version of the paper is not ready for publication at this conference.

**Reviewer Concerns:**

**Concerns that have been addressed**

1. The authors have clarified the rationality behind the selection of the performance metrics and the benchmarks.

2. The rebuttal has clearly answered the role of experimental Figures 2 and 3.

3. The authors have answered the risk of domination of high-dimensional covariates compared to the single-dimensional treatment variable.

---

**Concerns that remain partially or not fully resolved**

1. For the major concern of Reviewer 3g5t, although the authors give a concrete explanation for the equivalence between IPW and randomized adjustment, they still do not clarify the difference between $P(X,Y,Z)$ and $P(X)P(Z)P(X,Z)$, where the former is the empirical distribution from which the observed outcomes are drawn, while the latter is the focus of IWDD.

2. Similarly, for W2 raised by Reviewer uGJs, the rebuttal lacks an explicit intuitive answer, i.e., why the randomized adjustment can still work even if it has not seen $p(z)p(x)$ during training, except for the results from Lemma 1.

3. There is a need to further clarify the possibility of selecting other generative networks (e.g., GANs) as the backbone for W3 from Reviewer QBWv, rather than only illustrating the strength of diffusion models.

4. There is a lack of important generative baselines in the experiments, such as GANITE.

**Reviewer Scores:**

#### Reviewer Scores

1. Reviewer 3g5t may keep the score. **Reason:** The author do not fully address the questions/weaknesses raised by the Reviewer 3g5t.
2. Reviewer QBWv may keep the score. **Reason:** The author do not fully address the questions/weaknesses raised by the Reviewer QBWv.
3. Reviewer uGJs may keep the score. **Reason:** The author do not fully address the questions/weaknesses raised by the Reviewer uGJs.
4. Reviewer BX17 may raise the score to 5 or keep it the same. **Reason:** The author has partially solved the questions raised by the Reviewer, but some still remain unclear.

---

### Decision · Program_Chairs · 2026-01-26

Reject